# NEEDLE: A GENERATIVE AI-POWERED MULTI-MODAL DATABASE FOR ANSWERING COMPLEX NATURAL LANGUAGE QUERIES

## ABSTRACT

Multi-modal datasets, like those involving images, often miss the detailed descriptions that properly capture the rich information encoded in each item. This makes answering complex natural language queries a major challenge in this domain. In particular, unlike the traditional nearest neighbor search, where the tuples and the query are represented as points in a single metric space, these settings involve queries and tuples embedded in fundamentally different spaces, making the traditional query answering methods inapplicable. Existing literature addresses this challenge for image datasets through vector representations jointly trained on natural language and images. This technique, however, underperforms for complex queries due to various reasons. This paper takes a step towards addressing this challenge by introducing a Generative-based Monte Carlo method that utilizes foundation models to generate synthetic samples that capture the complexity of the natural language query and represent it in the same metric space as the multi-modal data. Following this method, we propose NEEDLE, a database for image data retrieval. Instead of relying on contrastive learning or metadata-searching approaches, our system is based on synthetic data generation to capture the complexities of natural language queries. Our system is open-source and ready for deployment, designed to be easily adopted by researchers and developers. The comprehensive experiments on various benchmark datasets verify that this system significantly outperforms state-of-the-art text-to-image retrieval methods in the literature. Any foundation model and embedder can be readily integrated into NEEDLE to improve its performance.

## 1 INTRODUCTION

Multi-modal datasets, like images, pose new challenges for data management systems. Unlike traditional databases, where tuple attributes are explicitly specified, these datasets often miss proper descriptions that capture the rich information encoded in each item. As a result, traditional query-answering approaches are not directly applicable in these settings. Meanwhile, modern personal devices like smartphones have made it possible to collect large amounts of multi-modal data, even from individual users. Due to the wealth of information "hidden" in each multi-modal tuple, the users may compose *complex natural language queries*[1], searching for the tuple(s) they are interested in. To further motivate this, let us consider the following running example.

EXAMPLE 1. A photo enthusiast has collected a large pool of images over the past several years in her private data repository. She wants to retrieve a specific photo she has in mind. She describes the picture as [A banana gazing at its reflection in a mirror].[2]

Traditional Nearest Neighbor (NN) search algorithms are not directly applicable here since they consider the dataset tuples and the query as points in the same data space. In contrast, in Example 1, (i) each tuple is an image, and (ii) the query is in the form of an unstructured, complex natural language statement. **Contrastive learning** methods are the state-of-the-art approaches for answering natural

---

[1]In this context, by "complex queries," we refer to retrieval tasks that require understanding compositionality, detailed attribute descriptions, and spatial relationships (e.g., "A blue napkin to the right of a white plate"), distinct from open-ended reasoning or Visual Question Answering (VQA) tasks.

[2]See Figure 12 for more examples.

language queries on image datasets, that train a vector representation (aka embedding) jointly on hundreds of millions of ⟨image, text⟩ pairs. Using a jointly trained embedding, one can transform the query and the dataset images to vector representations in the same space and apply NN-search to answer the query. This approach works satisfactorily for "simple" and "common" queries, such as simple object detection tasks.

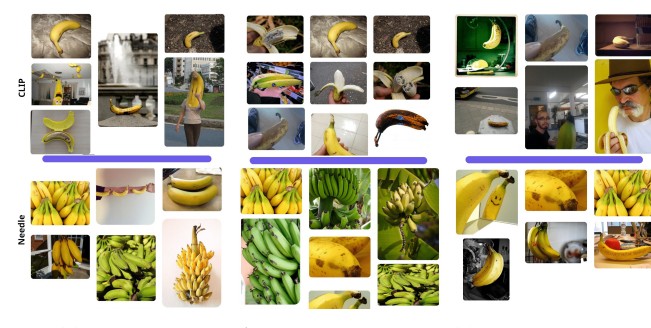

As a concrete example, continuing with our running example (Example 1), we used CLIP[3], the popular contrastive learning embedding introduced by OpenAI [51], to answer the natural language query [A banana] on the benchmark image dataset LVIS [22]. As reflected in the top-left cell of Figure 1, the results are satisfactory, as most of the returned images include a banana. However, it fails for a moderately complex query, [An unripe banana], as none of the returned images shows an unripe (green) banana – the top-center cell of Figure 1. It also fails on more complex queries, such as the one in Example 1, and does not retrieve the target image (the top-right cell of Figure 1). We observed a similar behavior to this example across a wide range of queries (§ 5).

(a) [a banana]  (b) [an unripe banana]  (c) [a banana gazing at its reflection in a mirror]

Figure 1: The query results for the queries in Example 1 using NEEDLE vs. CLIP [51].

This motivates us to take a step toward filling this gap by proposing a Monte Carlo randomized algorithm. In particular, we empower our algorithm with generative AI (GenAI) to generate a set of synthetic multi-modal tuples that represent the complex natural language query. Subsequently, leveraging a collection of embedders, we conduct a series of NN searches using the generated tuples and aggregate the results to generate the final query outputs. We use our method to develop NEEDLE[4], our deployment-ready system[5] for complex natural query answering on image datasets.

Unlike existing approaches, NEEDLE can handle natural language queries with different levels of complexity. To demonstrate this, let us consider our running example, Example 1 once again. We issued the same set of simple, moderate, and complex queries to NEEDLE. The results are provided in the second row of Figure 1. For the simple object-finding task, the bottom-left cell in Figure 1, we observed a similar (but slightly better) performance since all returned images satisfied the query, i.e., they included a banana. It also performed reasonably well for the moderately complex query, as a large portion of the output images were green bananas – the bottom-center cell in Figure 1. But, perhaps more interestingly, it could successfully find the image-of-interest described in the complex query of Example 1 – the bottom-right cell in Figure 1. Interestingly, the target image was returned as *the top-1 search result* (the top-left image in the bottom-right cell), demonstrating the ability of our system to answer these types of queries. We observed a consistent behavior for our system in experiments across a large number of different queries (§ 5).

**Related Work:** Our work intersects with several evolving domains. In **Multi-modal Data Retrieval**, approaches have progressed from contrastive learning models like CLIP [51] and ALIGN [25] to recent MLLM-based embeddings such as SigLIP [79] and E5-V [27]. Simultaneously, the emerging field of **Generative Information Retrieval** explores leveraging generative models to bridge modality gaps, as seen in concurrent works like IRGen [82] and SceneDiff [83]. Furthermore, our system draws upon **GenAI in Data Management**, where foundation models are increasingly utilized for data augmentation [57, 16] and automated query generation [63]. A comprehensive review of these areas is provided in Appendix B.

---

[3]We used ALIGN [25], FLAVA [58], and CoCa [78] as well as recent VLM-based baselines SigLIP [79] and E5-V [27] as additional baselines in our experiments.

[4]The name NEEDLE is inspired by the common expression "finding NEEDLE in the haystack," reflecting the challenge of pinpointing relevant information in large multimodal spaces.

[5]NEEDLE GitHub Repository: https://anonymous.4open.science/r/Needle-FFB4/

**Technical contributions:** This paper introduces a new approach to using GenAI for complex natural-language query answering on multi-modal data. We propose NEEDLE, a system for image data retrieval. In summary, our contributions are as follows:

1. **Theoretical Foundation:** We introduce a novel Monte Carlo method for answering complex natural-language queries on multi-modal data. We leverage Foundation models to generate synthetic tuples representing the queries. We then use a collection of embeddings that allows us to apply traditional k-NN techniques to retrieve related tuples. Our algorithm aggregates the output generated for each (synthetic tuple, embedding) pair to obtain the final result. We further analyze our estimator's properties, extending our theoretical bounds to account for correlations among embedders. (§ 2, Appendix C)

2. **System Design:** We present NEEDLE, an interactive system built upon our Monte Carlo method. Developed with production readiness in mind, NEEDLE emphasizes ease of deployment, efficiency, robustness, and user-friendly interaction via a command-line interface (§ 3, Appendix G).

3. **Efficiency Optimizations:** We propose and integrate several practical optimizations to transform NEEDLE into a robust and efficient system. To ensure high retrieval accuracy and robustness, we introduce a dynamic embedder trust mechanism to manage varying embedder reliability and an outlier detection method for removing poor-quality guide images (§ 3). To achieve high performance and low latency, we design a multi-stage inference pipeline featuring a query complexity classifier for short-circuiting simple queries and a novel caching mechanism that generates metadata from query history, creating a self-improving feedback loop (§ 4).

4. **Experimental Evaluation:** We conduct comprehensive experiments demonstrating NEEDLE's effectiveness across diverse benchmarks (including object detection and complex natural language queries) compared to state-of-the-art baselines, supported by ablation studies on sample variance and foundation model bias, and a user study indicating strong user preference for NEEDLE (§ 5, Appendix H).

**Remarks on Novelty:** During the review period, we became aware of concurrent independent works, subsequent to our technical report, which also explored some aspects of generative models for retrieval. However, existing methods often lack the robust probabilistic framework required to handle the stochastic nature of generation or fail to address the latency bottlenecks inherent to deployment. We distinguish NEEDLE from these approaches by introducing a Generative-based Monte Carlo framework and specific system optimizations, as further discussed in Appendix B.

## 2 THEORETICAL FOUNDATION: GENAI-POWERED MONTE-CARLO METHOD

The core challenge in answering a natural language query $\varphi$ over a multi-modal dataset $\mathcal{D}$ is bridging the semantic gap between the two different data types. While standard nearest-neighbor (NN) search over vector embeddings is effective when queries and data share a space, existing methods that create a joint embedding space for text and images often fail on complex queries [48, 29]. Our work introduces a new approach to solve this problem by leveraging Generative AI to transform the query itself into the multi-modal data space.

**Query Transformation and Randomized Algorithm:** Our method reframes the text-to-image search problem as an image-to-image search problem. The key idea is to use a foundation model (e.g., DALL·E 3) [80, 55, 52] to generate a set of $m$ synthetic **guide tuples**[6], $\{\bar{\mathbf{g}}_1, \cdots, \bar{\mathbf{g}}_m\}$, that act as stochastic representations of the query $\varphi$ in the image space. Each guide tuple $\bar{\mathbf{g}}_j$ is considered an i.i.d. sample from a distribution whose mean is the ideal (but unknown) tuple $\mathbf{g}_\varphi$ that perfectly represents the query. We acknowledge that in practice, foundation models may exhibit biases (e.g., missing compositional relations), which we mitigate by aggregating across diverse models (see § 5 for variance reduction analysis). Please refer to Appendix C for details about terms, notations, and preliminary discussions about our algorithm.

To account for variances in how different models interpret semantic similarity, we use an ensemble of $l$ distinctly-learned embedders, $\{\mathcal{E}^1, \cdots, \mathcal{E}^l\}$. We estimate the distance $\bar{\delta}_{\varphi,i}$ between the query $\varphi$ and any tuple $t_i$ in the dataset by averaging the distances $\delta(\ )$ between the guide tuples and $t_i$ across all embedders:

$$\bar{\delta}_{\varphi,i} = \frac{1}{m\,l} \sum_{j=1}^{m} \sum_{\ell=1}^{l} \delta\left(\mathcal{E}^\ell(\bar{\mathbf{g}}_j), \vec{v}_i^\ell\right) \tag{1}$$

where $\vec{v}_i^\ell$ is the vector representation of $t_i$ on embedder $\mathcal{E}^\ell$. The following theorem provides a probabilistic guarantee that our estimated distance $\bar{\delta}_{\varphi,i}$ is close to the optimal distance $\delta_{\varphi,i}$ under the assumption of independence.

---

[6]For image databases, each tuple is an image.

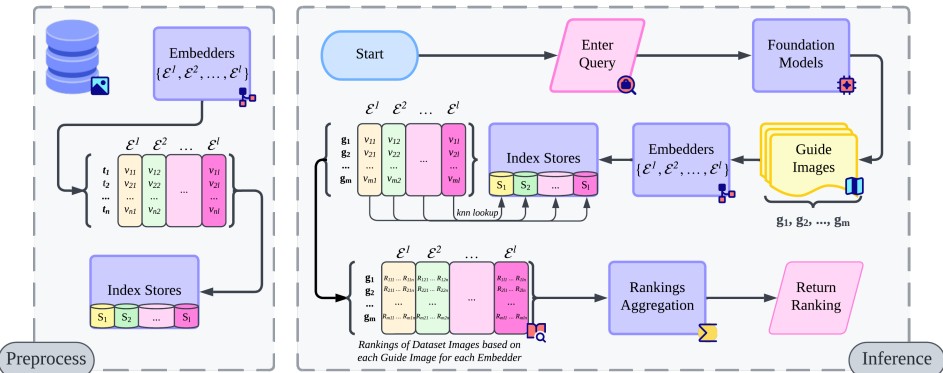

Figure 2: NEEDLE Architecture

**Theorem 1** [7] *For every tuple $t_i$ in a multi-modal dataset $\mathcal{D}$, let $\delta_{\varphi,i}$, be the true distance of $t_i$ to $\varphi$. Given a positive value $\gamma$, and assuming embedders provide independent observations,*

$$\Pr\left(\left(\frac{\bar{\delta}_{\varphi,i}}{\delta_{\varphi,i}} \geq (1+\gamma)\right) \vee \left(\frac{\bar{\delta}_{\varphi,i}}{\delta_{\varphi,i}} \leq (1-\gamma)\right)\right) \leq \mathbf{e}^{\frac{-m\,l\,\gamma^2\,\delta_{\varphi,i}}{3}} + \mathbf{e}^{\frac{-m\,l\,\gamma^2\,\delta_{\varphi,i}}{2}}.$$

**Correlated Embedders.:** The bound in Theorem 1 assumes that the $l$ embedders provide independent observations. However, distinct embedders may exhibit correlations due to shared architectural biases or training data. Following survey-sampling theory [70, 71], if the average pairwise correlation between embedders is $\rho$, the effective number of embedders is $l_{\text{eff}} = \frac{l}{1+(l-1)\rho}$. In this setting, the error bound in Theorem 1 holds by replacing $l$ with $l_{\text{eff}}$ in the exponents, resulting in a slightly weaker bound.

## 3 SYSTEM DESIGN: THE INITIAL PROTOTYPE

Building upon the theoretical foundations of our method, we designed an initial prototype of NEE-DLE for image retrieval. The architecture (Figure 2) consists of two main components. The first is a one-time **Preprocessing** phase, where the image dataset is indexed into a vector store using a set of predefined embedders. We use `Milvus` with an `HNSW` index for efficient approximate nearest-neighbor search. The second component is the **Inference Pipeline**. At query time, the system generates $m$ synthetic guide images using foundation models, transforms them into vector representations, performs a k-NN lookup for each, and aggregates the results.

While this direct application of our Monte Carlo method is effective, it introduces two significant practical challenges: firstly, how to effectively aggregate rankings when embedders exhibit varying performance across different query topics, and secondly, how to manage anomalous or low-quality guide images that can degrade retrieval accuracy. To address these challenges, our prototype incorporates two key mechanisms. The first is a dynamic **Embedder Trust Mechanism** to handle varying reliability. Recognizing that an embedder's performance is topic-dependent (e.g., one may excel at identifying animals but not vehicles), this mechanism assigns topic-specific reliability scores to each embedder. The final ranking is then produced via a weighted aggregation of each embedder's results based on these scores, which can be dynamically adjusted using user feedback via the Multiplicative Weight Update Method (MWUM). The second mechanism is an **Anomaly Detection** module to filter poor-quality guide images. This process first uses UMAP[45] to reduce the dimensionality of the guide image embeddings from all available embedders. Subsequently, it applies the Local Outlier Factor (LOF) algorithm to identify outliers based on neighborhood density. We emphasize that this mechanism is specifically tuned to filter out technical failures (e.g., solid black images triggered by safety filters or pure noise) rather than limiting stylistic diversity. This ensures that unique or creative guide images, which are valid representations of the query, are preserved to maintain the richness of the Monte Carlo sample. A guide image is flagged as anomalous and removed if its weighted-aggregate outlier score surpasses a set threshold.

The detailed algorithms and equations for these two mechanisms are provided in Appendix E. While this prototype demonstrates high retrieval accuracy (§ 5.2), it suffers from significant latency due to on-the-fly image generation. This limitation motivates the need for the efficiency enhancements detailed in the next section.

---

[7]The proof is provided in Appendix D.

## 4 SYSTEM OPTIMIZATION: ADDRESSING THE EFFICIENCY ISSUE

While the initial prototype demonstrates superior retrieval accuracy, its reliance on on-the-fly image generation introduces substantial latency. This section details the enhancements implemented to transform NEEDLE into a highly efficient system without compromising accuracy. Our approach follows two primary strategies: first, minimizing the frequency of expensive image generation, and second, making the generation pipeline itself faster when it is unavoidable.

To minimize image generation, we introduce a synergistic system built on two core components. The first is a **Query Complexity Classifier** that predicts whether a query is "simple" enough to be handled accurately by fast, existing contrastive learning based methods. If so, the system **short-circuits** the pipeline, bypassing image generation entirely. The second component is an **Implicit Metadata Generation** mechanism, which creates a positive feedback loop. For complex queries that run through the full pipeline, the system uses the query text to tag the high-confidence results. This continually enriches the dataset's metadata, making the preliminary text-based search used by the classifier more powerful over time. As the system processes more queries, it gets better at identifying simple ones, progressively reducing its reliance on the generation pipeline.

For complex queries that still require image generation, we developed an **Optimized Generation Pipeline**. This pipeline is heavily optimized for speed based on insights from our experimental analysis; it employs fast open-source foundation models like SD-Turbo for rapid generation, uses a minimal configuration of a single guide image at $512 \times 512$ resolution (which we demonstrate in § 5 is sufficient for high accuracy), and utilizes a small set of highly efficient embedders such as EVA and RegNet, as robust performance is achievable with only a few strong models.

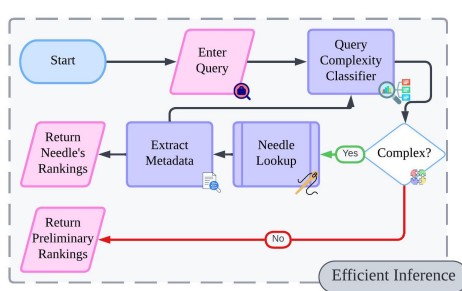

Figure 3: The efficiency-first pipeline.

### 4.1 AN OVERVIEW OF THE ENHANCED PIPELINE

In summary, the enhanced NEEDLE processes each query through the multi-stage, efficiency-first pipeline illustrated in Figure 3. This process begins with the **Query Complexity Classifier**, which short-circuits the pipeline for simple queries by immediately returning results from preliminary methods. If a query is deemed complex, it proceeds to the **Optimized NEEDLE Pipeline**, which uses the efficient guide image generation configuration. The process concludes with the **Metadata Tagging** module, where high-probability results are tagged with the query text. This enriches the metadata, improving future classifier performance and completing the self-improvement loop.

This architecture ensures that the computationally expensive generation process is used only when necessary, while creating a system that learns from user queries to become more efficient over time. The technical details for the classifier and metadata generation are provided in Appendix F.

### 4.2 SYSTEM IMPLEMENTATION AND DEPLOYMENT

Building on these efficiency enhancements, we developed NEEDLE as a complete, open-source database designed for production environments. Our primary goal was to create a practical and user-friendly system that researchers and developers can easily adopt. Key design principles included a powerful command-line interface (CLI) for seamless user interaction, a modular microservice architecture to easily integrate new embedders and foundation models, and containerized deployment for robust, cross-platform compatibility. A comprehensive overview of the system's development objectives, architecture, technical stack, and installation is provided in Appendix G.

## 5 EXPERIMENTAL EVALUATION

In this section, we evaluate several key aspects of NEEDLE to demonstrate its capabilities and to evaluate its performance. We use multiple benchmark datasets and several baselines for this purpose. In the following, we first detail our experiments setup (§ 5.1), followed by a proof-of-concept analysis (§ 5.2) that demonstrates NEEDLE's efficacy in text-to-image retrieval across diverse object detection and complex natural language query benchmarks. Subsequently, we conduct an ablation study in Section 5.3 to assess the impact of hyperparameter variations on NEEDLE's performance. Finally, in Appendix H.5 we provide an end-to-end case study, involving human participants' qualitative feedback on NEEDLE's responsiveness to arbitrary user-generated queries.

Table 1: Zero-shot retrieval performance on object detection datasets. For each dataset, we report R@10, P@10, MAP, and MRR. Each cell shows two numbers (All / Hard).

| | Caltech256 | | | | COCO | | | | LVIS | | | | BDD | | | |
|---|---|---|---|---|---|---|---|---|---|---|---|---|---|---|---|---|
| Baseline | R@10 | P@10 | MAP | MRR | R@10 | P@10 | MAP | MRR | R@10 | P@10 | MAP | MRR | R@10 | P@10 | MAP | MRR |
| CLIP | 0.926 | 0.926 | 0.939 | 0.952 | 0.934 | 0.934 | 0.952 | 0.988 | 0.177 | 0.167 | 0.168 | 0.316 | 0.660 | 0.660 | 0.670 | 0.714 |
| | 0.150 | 0.150 | 0.181 | 0.193 | 0.400 | 0.400 | 0.477 | **1.000** | 0.093 | 0.083 | 0.078 | 0.224 | 0.033 | 0.033 | 0.005 | 0.048 |
| ALIGN | 0.941 | 0.941 | 0.947 | 0.961 | 0.944 | 0.944 | 0.960 | 0.981 | 0.215 | 0.201 | 0.207 | 0.379 | 0.560 | 0.560 | 0.573 | 0.704 |
| | 0.375 | 0.375 | 0.398 | 0.541 | 0.800 | 0.800 | 0.895 | **1.000** | 0.145 | 0.130 | 0.129 | 0.306 | 0.000 | 0.000 | 0.003 | 0.014 |
| FLAVA | 0.882 | 0.882 | 0.903 | 0.949 | 0.924 | 0.924 | 0.941 | 0.963 | 0.185 | 0.172 | 0.180 | 0.321 | 0.660 | 0.660 | 0.698 | 0.725 |
| | 0.258 | 0.258 | 0.306 | 0.491 | 0.100 | 0.100 | 0.281 | **1.000** | 0.109 | 0.097 | 0.099 | 0.241 | 0.033 | 0.033 | 0.036 | 0.083 |
| BLIP + MiniLM | 0.826 | 0.826 | 0.838 | 0.880 | 0.941 | 0.941 | 0.951 | 0.975 | 0.180 | 0.177 | 0.179 | 0.332 | 0.600 | 0.600 | 0.610 | 0.725 |
| | 0.317 | 0.317 | 0.372 | 0.408 | 0.700 | 0.700 | 0.698 | **1.000** | 0.115 | 0.111 | 0.107 | 0.260 | **0.167** | **0.167** | 0.144 | **0.333** |
| PlugIR | 0.889 | 0.889 | 0.934 | 0.936 | 0.927 | 0.927 | 0.949 | 0.983 | 0.189 | 0.172 | 0.173 | 0.326 | 0.600 | 0.600 | 0.678 | 0.731 |
| | 0.225 | 0.225 | 0.391 | 0.523 | 0.700 | 0.700 | 0.654 | **1.000** | 0.107 | 0.091 | 0.112 | 0.276 | 0.033 | 0.033 | 0.074 | 0.245 |
| CoCa | 0.780 | 0.780 | 0.818 | 0.880 | 0.933 | 0.933 | 0.951 | 0.951 | 0.192 | 0.184 | 0.187 | 0.354 | 0.560 | 0.560 | 0.583 | 0.713 |
| | 0.075 | 0.075 | 0.094 | 0.118 | 0.700 | 0.700 | 0.642 | **1.000** | 0.131 | 0.116 | 0.128 | 0.272 | 0.033 | 0.033 | 0.003 | 0.102 |
| SigLIP | 0.951 | 0.951 | 0.965 | 0.956 | [RES] | [RES] | [RES] | [RES] | 0.310 | 0.277 | 0.311 | 0.492 | 0.660 | 0.660 | 0.659 | 0.736 |
| | 0.569 | 0.569 | 0.572 | 0.669 | [RES] | [RES] | [RES] | [RES] | 0.256 | 0.212 | 0.213 | 0.403 | 0.050 | 0.050 | 0.067 | 0.143 |
| E5-V | 0.926 | 0.926 | 0.923 | 0.929 | [RES] | [RES] | [RES] | [RES] | 0.318 | 0.289 | **0.327** | 0.502 | 0.744 | 0.744 | 0.725 | 0.786 |
| | 0.243 | 0.243 | 0.359 | 0.257 | [RES] | [RES] | [RES] | [RES] | 0.259 | 0.218 | 0.223 | 0.398 | 0.167 | 0.167 | 0.125 | 0.323 |
| Needle | **0.962** | **0.962** | **0.966** | **0.979** | **0.966** | **0.966** | **0.977** | **1.000** | **0.330** | **0.295** | 0.323 | **0.511** | 0.720 | 0.720 | 0.711 | 0.750 |
| | **0.667** | **0.667** | **0.687** | **0.776** | **0.900** | **0.900** | **0.981** | **1.000** | **0.263** | **0.225** | **0.249** | **0.453** | 0.167 | 0.167 | **0.158** | **0.333** |

## 5.1 EXPERIMENTS SETUP

**Baselines:** We evaluate our method using several prominent vision-language models as baselines, representing different architectural and training paradigms. These include: well-established contrastive learning models such as CLIP [51] and ALIGN [25]; unified models that combine contrastive and generative objectives like FLAVA [58] and CoCa [78]; an effective decoupled pipeline combining BLIP for captioning with MiniLM for text embedding [37, 69]; and PlugIR [33], a framework that uses large language models for retrieval. Additionally, to compare against the most recent advancements in multimodal embeddings, we include SigLIP [79] and E5-V [27]. Each of these models offers a unique perspective on learning joint image-text representations, and together they provide a comprehensive set of comparisons for our work. Detailed descriptions of these models and their specific configurations used in our experiments are deferred to Appendix H.1.

**Datasets:** We evaluate NEEDLE using a comprehensive set of datasets, categorized into those focusing on object detection and those involving complex natural language queries. The object detection benchmarks include Caltech256 [21], MS COCO [39], LVIS [22], and BDD100k [77]. For complex natural language queries, we utilize COLA [54], Winoground [61], NoCaps [3], and SentiCap [44]. Detailed descriptions of the datasets are provided in Appendix H.2.

**Evaluation Metrics:** We use distinct evaluation metrics tailored to the characteristics of each benchmark type. For object detection benchmarks, we measure performance using Mean Recall at $k$ (R@k), Precision at $k$ (P@k), Mean Average Precision (MAP), and Mean Reciprocal Rank (MRR). For complex natural language query experiments, we report both MRR and Pairing Accuracy (PAcc). Detailed definitions and formulas are provided in Appendix H.3.

**Embedders and Hardware Configuration:** We use all or a subset of embedders listed in the Appendix (Table 6) in our experiments. All experiments were conducted on two servers, each equipped with 32 GB of memory, a 12-core CPU, and two Tesla T4 GPUs, running Ubuntu 22.04 LTS.

**Foundation Models and the Monetary Cost:** For image generation in our experiments, we employed several state-of-the-art foundation models, namely `DALL-E 3` [47], `Imagen 3` [20], `Flux-Schnell` [9], and `RealvisXL-v3.0-turbo` [2]. The total monetary cost incurred for utilizing these models was $257.53 (USD).

## 5.2 PROOF OF CONCEPT

**Task 1. Object Detection:** We begin our experiments by evaluating the performance of NEEDLE against the baselines on object detection benchmarks, utilizing the Caltech256, COCO, LVIS, and BDD datasets. For each object in these benchmarks, we formulate queries to retrieve them. For NEEDLE, we employed three image generation engines (`Flux-Schnell`, `RealVisV3`, and `ImagenV3-fast`) to generate three images per engine, resulting in a total of nine guide images per query, all at MEDIUM quality. Additionally, we utilized the ensemble of embedders detailed in Table 6. Following the methodology outlined in Appendix H.1, we considered queries with a CLIP AP below 0.5 as the "hard set". Table 1 presents the performance of NEEDLE against the baselines across R@10, P@10, MAP, and MRR metrics. Notably, NEEDLE demonstrates superior performance compared to all baselines on both the easy and hard sets. Specifically, for the hard set, NEEDLE achieves MAP improvements of 73%, 10%, 93%, and 10% over the second-best baseline

Table 2: Zero-shot retrieval performance on complex natural language queries benchmarks.

|  | Cola | | Winoground | | SentiCap | NoCaps |
|---|---|---|---|---|---|---|
| Baseline | PAcc | MRR | PAcc | MRR | MRR | MRR |
| CLIP | 0.578 | 0.246 | 0.519 | 0.426 | 0.464 | 0.573 |
| ALIGN | 0.591 | 0.301 | 0.554 | **0.501** | 0.555 | 0.704 |
| FLAVA | 0.615 | 0.336 | 0.574 | 0.482 | 0.546 | 0.658 |
| BLIP + MiniLM | 0.449 | 0.195 | 0.485 | 0.330 | 0.331 | 0.398 |
| Needle | **0.631** | **0.352** | **0.593** | 0.490 | **0.642** | **0.745** |

Table 3: Inference Time Breakdown for optimized NEEDLE on LVIS

| Component | Time (seconds) |
|---|---|
| Image Generation | 0.136 |
| Embedder 1 (RegNet) | 0.021 |
| Embedder 2 (EVA) | 0.030 |
| Retrieval (Embedder 1) | 0.003 |
| Retrieval (Embedder 2) | 0.002 |
| Ranking Aggregation | 0.0001 |
| **Total (with overheads)** | **0.203** |

(a) BDD  (b) Caltech256  (c) COCO  (d) LVIS

Figure 4: Impact of the number of guide images and embedders on NEEDLE's MAP.

on Caltech256, COCO, LVIS, and BDD datasets, respectively. These results underscore the substantial advantage of using synthetic data for image retrieval over traditional contrastive learning and image-to-text approaches. Furthermore, Figure 12 provides a visual illustration of the performance of NEEDLE (v.s. CLIP) and the synthetic guide images generated for each query.

**Task 2. Complex Natural Language Queries:** To evaluate the system's ability to handle challenging language and visual nuances, we conduct experiments across two distinct retrieval tasks. The first, a *Pairing Task*, assesses the capability to correctly match a caption to its corresponding image when presented with two highly similar images and captions that differ only in subtle details. Evaluated on the Winoground [61] and COLA [54] datasets (see Figure 11), performance is measured by Pairing Accuracy (PAcc), where random chance is 0.25. The second, a *Full-set Retrieval Task*, measures performance when retrieving a specific image from an entire dataset using a single, complex query. For this task, we report the Mean Reciprocal Rank (MRR) on the Winoground, COLA, SentiCap [44], and NoCaps [3], which all feature challenging captions requiring a nuanced understanding for successful retrieval from a large pool.

Table 2 presents the Pairing Accuracy and Mean Reciprocal Rank (MRR) metrics for different aforementioned baselines. As illustrated in the table, NEEDLE outperforms the baselines in both PAcc and MRR. It is important to note that current foundation models still exhibit limitations in generating images that are fully aligned with an input query. In particular, our investigation reveals that these models frequently fail to produce images with the correct compositional ordering. We anticipate that further advancements in foundation models will further enlarge the performance gap between NEEDLE and the baselines. It is also noteworthy that BLIP+MiniLM performed significantly worse than the other baselines. Initially, we anticipated that this baseline would serve as a strong competitor in handling complex natural language queries, given its design to capture compositional and relational attributes between objects in images; however, the empirical results did not corroborate these expectations. In contrast, FLAVA demonstrated exceptionally strong performance in these tests.

### 5.3 HYPER-PARAMETER ABLATION STUDY

In this section, we examine the impact of each hyper-parameter on NEEDLE's performance.

**Impact of Guide Image and Embedder Counts:** To assess the sensitivity of NEEDLE's performance to its core hyperparameters, we evaluate the impact of varying the number of guide images generated per engine ($m$) and the number of embedders used ($\ell$). We conducted experiments on the BDD, Caltech256, COCO, and LVIS datasets with $m \in \{1, 2, 3\}$ and $\ell \in \{1, 2, 4, 6\}$, adding embedders in descending order of their reliability weights. As illustrated in Figure 4, Mean Average Precision (MAP) generally improves as both $m$ and $\ell$ increase, which suggests that achieving a consensus among multiple embedders enhances retrieval reliability. Notably, even the minimal configuration ($m = 1, \ell = 1$) outperforms most baselines (Table 1), demonstrating the fundamental strength of our approach. Furthermore, the analysis reveals a pattern of diminishing returns; for instance, the performance gain from increasing $m$ from 1 to 2 is substantially larger than from 2 to 3. This finding indicates that NEEDLE can achieve high accuracy with a small number of guide images and embedders, reinforcing the efficiency of its design without requiring extensive image generation.

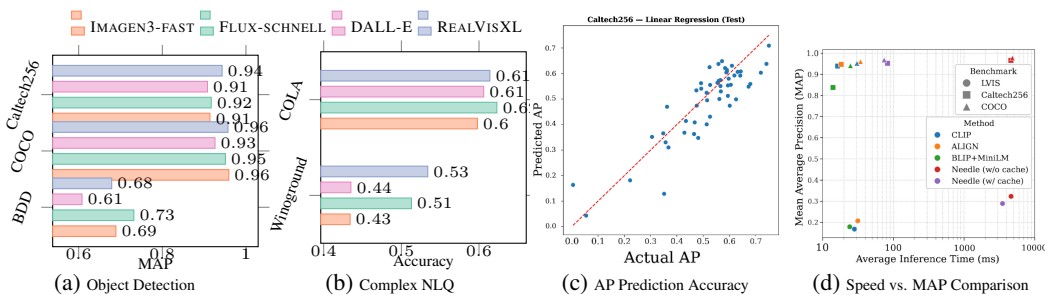

Figure 6: Foundation Model Performance and Query Complexity Analysis: (a) Object Detection benchmarks, (b) Complex Natural Language Query benchmarks, (c) Query Complexity Classifier AP prediction accuracy, (d) Inference speed vs. MAP comparison across baselines and benchmarks.

**Sample Variance Analysis:** We further investigated the effect of guide images and embedders on the stability of our estimator. As the number of guide images ($m$) and embedders ($l$) increases, the variance of the estimated distance $\bar{\delta}_{\varphi,i}$ is expected to decrease, leading to more robust retrieval. Figure 5 presents the empirical standard error of the distance estimator as a function of $m$ and $l$. The results confirm that our Monte Carlo ensembling strategy effectively smooths out the stochasticity of individual guide generations and embedder biases, resulting in a consistent estimator even with a modest number of samples.

**Foundation Model Analysis:** NEEDLE is designed to be model-agnostic, supporting various open-source and proprietary foundation models for guide image generation. To analyze this flexibility, we assess the individual performance of different models and evaluate the benefit of ensembling them. Our evaluation utilized four models—the proprietary DALL-E3 and IMAGEN3-FAST, and the open-source FLUX-SCHNELL and REALVISXL. We benchmarked their performance on object detection

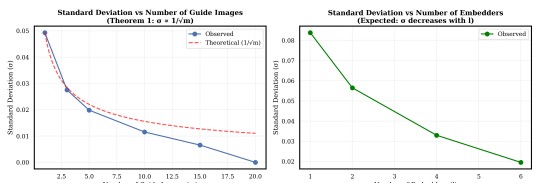

Figure 5: Reduction in estimator sample variance as a function of the number of guide images ($m$) and embedders ($l$).

tasks (BDD, Caltech256, COCO) using Mean Average Precision (MAP) and on complex natural language queries (Winoground, COLA) using Accuracy. The latter task was included specifically to test the models' ability to handle nuanced prompts where alignment and image quality are critical.

The individual performance of each foundation model is presented in Figure 6. A key observation is that the open-source models, FLUX-SCHNELL and REALVISXL, substantially outperformed the proprietary models on complex queries, suggesting a superior capability for prompt alignment and compositional understanding. Among the models, DALL-E3 generally yielded lower performance, while REALVISXL proved to be a consistently strong performer across most benchmarks.

We also analyzed the specific biases introduced by these models. Our qualitative observations suggest two primary failure modes: a "bag-of-words" bias where models ignore strict compositional ordering (e.g., placing an object to the left vs. right), and an abstraction bias where abstract concepts (e.g., "sadness") are grounded into stereotypical visual tropes (e.g., rain). However, as discussed below, ensembling helps mitigate these individual failures.

Beyond individual model performance, we analyzed the effect of ensembling multiple foundation models. As shown in Figure 7a, increasing the number of models in the ensemble consistently improves NEEDLE's overall performance. This benefit is more pronounced for complex natural language queries than for object detection. The outcome is likely attributable to the increased probability of generating an image that accurately captures the nuanced details of a complex prompt, thereby facilitating correct identification. In contrast, for simpler object detection tasks, the primary benefit of an ensemble is an increase in recall due to the diverse representations of the target object across various formats and styles.

**Image Quality:** Foundation models usually support generating images in different resolutions. In this ablation study, we investigate the effect of image resolution (size) of guide images on NEEDLE's performance. We define three supported image sizes for each foundation model, SMALL, MEDIUM and LARGE. The exact amount of pixels is determined based on the foundation model design. We generate 3 images per query and utilize Flux-Schnell, RealVisXL and DALL-E for

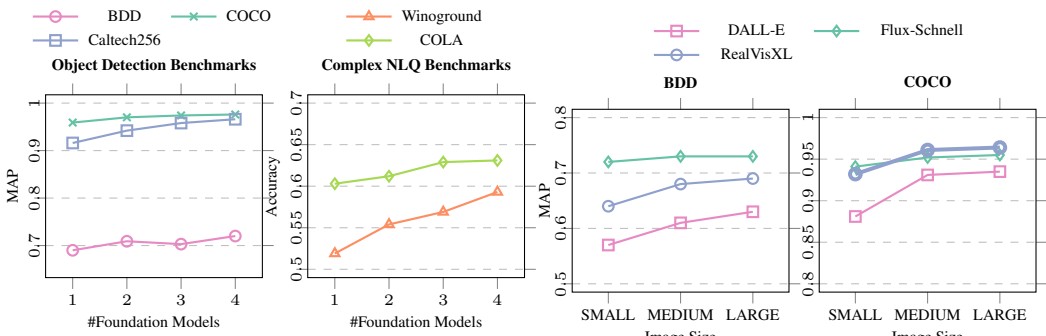

(a) The effect of number of Foundation Models on the Performance of NEEDLE for different Benchmarks

(b) Illustration of variation of NEEDLE MAP based on size of guide image for different foundation models

Figure 7: Foundation Models Analysis: (a) Effect of number of Foundation Models on Performance, (b) Variation of NEEDLE MAP based on size of guide image for different foundation models

this study (ImagenV3-fast does not support different images sizes as input). Figure 7b illustrates the MAP performance of various foundation models on the LVIS and COCO benchmarks across different image quality settings. It is evident that foundation models are typically fine-tuned for a specific image resolution, which in turn yields the best quality and prompt alignment. In our experiments, DALL-E demonstrated a consistent improvement in performance with increasing image sizes, whereas Flux-Schnell and RealVisXL exhibited a less pronounced correlation between performance and image size, particularly between the MEDIUM and LARGE settings.

**Outlier Detection Analysis:** While foundation models have shown remarkable capabilities in generating high-quality images, they are not infallible. Generated guide images may occasionally be semantically off-target or poorly aligned with the input query, which could negatively impact retrieval performance. To address this challenge, we implemented an outlier detection mechanism using the Local Outlier Factor (LOF) algorithm that identifies and filters out guide images that are semantically distant from the rest of the generated set before they are used for retrieval.

To evaluate the effectiveness of this mechanism, we conducted a comprehensive ablation study comparing three configurations: (1) no outlier detection, (2) human-evaluated filtering (where human annotators manually identified and removed poor-quality guide images), and (3) our automated LOF-based outlier detection. We evaluated these configurations on the BDD and LVIS benchmarks using RealVisXL as the foundation model.

Table 4 presents the results of the outlier detection analysis. The automated LOF-based outlier detection achieves performance that closely approximates human evaluation, demonstrating its effectiveness in identifying and filtering problematic guide images. Notably, even without any outlier detection, the generated images maintain sufficient quality for retrieval tasks, suggesting that current foundation models are generally reliable for this application. However, the consistent improvements observed with both human evaluation and automated detection confirm that outlier filtering provides meaningful performance gains, particularly on challenging benchmarks like LVIS where the long-tail distribution of categories makes retrieval more sensitive to guide image quality.

**Contribution Analysis: Generative Method vs. Embedder Ensemble:** To validate our design and disentangle the sources of NEEDLE's performance gains, we analyze the distinct contributions of its two primary components: the generative Monte Carlo framework and the use of an embedder ensemble. We conduct a controlled ablation study to isolate these effects. A key aspect of our analysis is measuring the improvement from the generative framework itself, independent of embedder choice. To do this, we compare the standard CLIP baseline against NEEDLE when constrained to use only the CLIP embedder. This comparison is critical because our method's core innovation—transforming text-to-image into image-to-image retrieval—unlocks the use of powerful image-only embedders (like EVA[18]) that are inaccessible to traditional approaches.

Table 4: Outlier Detection Ablation Study: MAP performance comparison across different filtering strategies on BDD and LVIS.

| Benchmark | No Detection | Human Eval. | LOF Detection |
|---|---|---|---|
| BDD | 0.68 | 0.70 | 0.73 |
| LVIS | 0.25 | 0.28 | 0.30 |

Table 5: Contribution Analysis: MAP performance comparison isolating the generative method's contribution using CLIP-only config.

| Benchmark | CLIP Base. | NEEDLE (CLIP) | NEEDLE (EVA) | NEEDLE (Full) |
|---|---|---|---|---|
| LVIS | 0.168 | 0.228 | 0.274 | 0.323 |
| Caltech256 | 0.939 | 0.951 | 0.958 | 0.966 |
| MS COCO | 0.952 | 0.961 | 0.970 | 0.977 |

The results of this decomposition, presented in Table 5, reveal a clear hierarchy of performance improvements. First, by comparing the CLIP baseline to NEEDLE (CLIP only), we isolate the gain from our generative method. This alone provides a substantial performance improvement, yielding a 35% relative gain in MAP on the LVIS dataset (from 0.168 to 0.228) and demonstrating the efficacy of using synthetic guide images. Second, the results show the benefit of unlocking stronger, image-only embedders, as seen in the performance increase from NEEDLE (CLIP only) to NEEDLE (EVA only). Finally, the move from a single strong embedder to the full ensemble provides further incremental gains, validating our ensembling strategy. This analysis clearly establishes that while the choice of embedder and the ensembling strategy are important contributors, the most significant performance gain originates from our novel generative framework.

**Computational Efficiency Analysis:** Assessing the computational efficiency of NEEDLE is essential for establishing its practical viability, particularly as its overhead stems from on-the-fly guide image generation and multi-embedder processing. To provide concrete performance metrics, we conducted a detailed timing analysis of NEEDLE's fully optimized configuration, which uses SD-Turbo with the efficient EVA and RegNet embedders. The evaluation was performed on the LVIS dataset (100k images) using 50 representative queries that require the full generation pipeline.

As shown in Table 3, the optimized pipeline's total inference time is 0.203s, a figure that is highly competitive with traditional baselines like CLIP (0.184s) while offering superior retrieval accuracy. We emphasize that this measurement relies on a configuration using a single guide image at $512 \times 512$ resolution, generated by SD-Turbo, which our ablation study (Figure 4) confirms is sufficient for high performance. It is important to note that this measurement represents the performance for complex queries that necessitate image generation. The average system-wide latency is expected to be substantially lower, as the Query Complexity Classifier and caching mechanism will bypass this pipeline entirely for simpler or repeated queries. Furthermore, NEEDLE exhibits favorable scalability characteristics; the primary computational cost, the image generation, is independent of the database size, making it well-suited for large-scale deployments. The inherently parallelizable nature of both the generation and embedding stages further enhances its practical applicability.

**Query Complexity Classifier Evaluation:** This section empirically evaluates the impact of our Query Complexity Classifier on NEEDLE's overall efficiency and retrieval performance. We observe that this module can effectively reduce computational overhead by dynamically routing queries, without a meaningful compromise in retrieval effectiveness. For these experiments, the classifier was implemented as a Linear Regression model trained to predict the expected Average Precision (AP) of a query when processed by preliminary retrieval methods. We derived features from two efficient baselines, CLIP [51] and ALIGN [25], including their top-$K$ mean cosine similarity scores, inter-method result overlap, and confidence deviation, as detailed in Appendix F.1. The model was trained on the combined training splits of the Caltech256 [21] and LVIS [22] benchmarks, using the actual AP from the preliminary methods as the ground-truth target. Figure 6c validates the classifier's effectiveness, showing a strong correlation between its predicted and actual AP scores.

The primary benefit of this classifier is a significant enhancement in NEEDLE's average inference speed. To quantify this, we compared the retrieval accuracy (MAP) and speed of NEEDLE —with and without the classifier—against established baselines across several benchmarks. As illustrated in Figure 6d, the classifier enables NEEDLE to achieve a much faster average inference time with only a negligible loss in MAP. This speedup is achieved by accurately identifying "simple" queries and short-circuiting the computationally intensive guide image generation process. Crucially, NEEDLE's efficiency is designed to improve over time. As the implicit metadata generation mechanism (§ F.2) populates the system with query-based tags, the effectiveness of the preliminary text-search method increases. This will naturally raise the number of queries classified as simple, further reducing reliance on on-the-fly image generation and progressively enhancing overall system speed.

## 6 CONCLUSION

This paper introduces a **Generative AI-powered Monte Carlo method** for answering complex text queries on multi-modal data. The technique uses AI to generate synthetic "guide" data that translates the query, enabling effective nearest-neighbor search. This method was implemented in an open-source image retrieval system called NEEDLE, which experiments show **significantly outperforms state-of-the-art** approaches. Future work aims to extend this technique to other data types, like audio and video, as the required AI models become available.

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

# APPENDIX

## A  DISCUSSIONS AND LIMITATIONS

**Reliance on Existing Models:** NEEDLE relies on Foundation Models for guide tuple generation and on embedders for generating vector representations of images, which may have their own capabilities, limitations, and inherent biases. This reliance presents both strengths and weaknesses in NEEDLE's architecture. The strength lies in its ability to improve performance by upgrading the internal Foundation Models and embedders. On the other hand, though, the limitations of these models also limit NEEDLE. Specifically, our experiments revealed two primary biases in current foundation models: (1) a "bag-of-words" bias, where models often ignore strict compositional or spatial instructions (e.g., placing object A specifically to the left of object B), and (2) an abstraction bias, where abstract concepts (e.g., "sadness") are frequently grounded into stereotypical visual tropes (e.g., rain or grayscale filters). To mitigate this, NEEDLE supports multiple Foundation Models and embedders, enabling it to draw from a broader range of knowledge and reduce the impact of individual model limitations via the ensemble-based Monte Carlo aggregation.

**Preprocessing Efficiency:** During the preprocessing phase, we employ a predefined set of embedders to generate vector representations of images. NEEDLE processes the images in parallel while utilizing the available GPU resources to distribute the workload efficiently. Moreover, NEEDLE supports multiple operational modes upon installation. For instance, the fast mode utilizes only two embedders, which offer preprocessing speeds that are comparable to those of state-of-the-art methods.

**Inference Efficiency:** NEEDLE's inference consists of three main sequential operations: (a) generating guide tuples, (b) k-NN lookup, and (c) ranking aggregation. Theoretically, for retrieving $k$ results from a dataset of size $n$ with constants $m$ (number of guide images) and $\ell$ (number of embedders), Step (a) is a constant-time operation. Step (b)'s time complexity depends on the underlying vector store algorithm, and in the current implementation, we leverage HNSW data structure index [43], which provides k-NN lookup time complexity of $O(k \log(n))$, while Step (c) uses Fagin's instance-optimal TA algorithm for rank aggregation [17].

In practice, however, the bottleneck lies in Step (a), generating guide tuples.

NEEDLE's current implementation supports both on-device and online image generation models. For optimal performance, we recommend using fast on-device generation models to eliminate network latency. Moreover, as demonstrated in our ablation study, NEEDLE maintains high accuracy even with lower-quality, lower-resolution images. Therefore, employing fast, lower-quality foundation models can significantly accelerate the overall process. In addition to these, the built-in Query Complexity Classifier (Appendix F.1) enables NEEDLE to generate images only when the query is deemed complex for preliminary methods, which drastically reduces the average inference time compared to using NEEDLE for all queries.

**Extensions to Other Modalities:** The core idea behind NEEDLE is adaptable to other modalities beyond images, including audio and video. However, due to the current limitations in publicly available foundation models, our focus in this paper was on the image data. As foundation models for these additional modalities continue to improve and become more accessible, we plan to extend NEEDLE to support them. This will allow NEEDLE to operate across multiple data types, unlocking further potential for handling complex, multi-modal queries in the near future.

**Sensitivity to Prompts and Random Seeds:** We analyzed the system's sensitivity to both random seed selection and prompt phrasing. Regarding seed robustness, as illustrated in our ablation study (Figure 4), retrieval performance stabilizes as the number of guide images ($m$) increases. By averaging over $m$ generated images, our Monte Carlo framework inherently smooths out the stochastic noise associated with specific random seeds. Regarding prompt sensitivity, we utilized the raw user query directly as the input for generation. Our preliminary experiments with prompt sanitization (e.g., removing parentheses, underscores, and special characters) yielded negligible performance improvements, confirming that modern foundation models are highly robust to minor syntactic variations. We acknowledge, however, that *semantic* prompt enhancement is a valuable direction for future work; specifically, leveraging LLMs to expand query details could potentially yield higher-quality guide tuples.

## B  RELATED WORK

Our paper relates to the following literature:

**Multi-modal Data Retrieval:** Multi-modal data retrieval (aka cross-modal retrieval), integrating various data types, has recently attracted significant attention. Early methods focused on shared representations by mapping modalities into a common latent space using canonical correlation analysis (CCA) [53, 68] and kernel-based techniques [26]. Recent approaches leverage deep learning models, including cross-modal encoders [66, 40] and joint embedding networks [7, 56]. Advanced techniques, such as attention mechanisms [74, 60] and contrastive learning [50], further enhance retrieval by capturing fine-grained relationships. CLIP [51] and ALIGN [25] are widely adopted methods based on contrastive learning. More recently, the field has evolved towards using Multi-modal Large Language Models (MLLMs) to generate universal embeddings, with state-of-the-art approaches such as SigLIP [79], E5-V [27], and VLM2Vec [28] achieving strong performance on massive embedding tasks.

**Generative Information Retrieval:** A burgeoning body of concurrent work has begun exploring the intersection of generative models and information retrieval. Approaches such as IRGen [82], SceneDiff [83], and Diffusion Augmented Retrieval [41] utilize generative models to synthesize visual features or query representations. Similarly, Generative Zero-Shot retrieval [67] and GE-NIUS [30] have investigated using generated content to bridge the modality gap.

**Remarks on Novelty:** While NEEDLE shares the conceptual motivation of using generative models with the aforementioned concurrent works, it distinguishes itself through its theoretical and system-oriented contributions. Unlike previous methods that often rely on single-model generation or specific adapter training, NEEDLE introduces a *Generative-based Monte Carlo method*. This framework rigorously treats generated tuples as probabilistic samples, employing an ensemble of foundation models and embedders to mitigate the bias and variance inherent in individual generators. Furthermore, NEEDLE goes beyond the algorithmic layer to address the practical latency bottlenecks of this paradigm, introducing system-level innovations such as the Query Complexity Classifier and implicit metadata caching to ensure deployment readiness.

**GenAI in Data Management:** The intersection of Generative AI (GenAI) and data management has emerged as a promising area, driving innovations in how data is generated, managed, and utilized. GenAI models, such as GPT [1] and Stable Diffusion [55], have been employed for synthetic data generation to augment datasets [59, 57], improve model training [32, 62], and address data scarcity or imbalance issues [16]. In data cleaning and integration, these models assist with tasks like imputation [65], and anomaly detection by generating plausible data patterns [75]. Recent works also explore the role of GenAI in automating query generation [63, 64, 38] and data summarization [14], enhancing the efficiency of database systems and analytics platforms.

**Natural Language Query Processing:** The evolution of natural language to structured query translation has advanced from rule-based systems to sophisticated large language models (LLMs). Early methods used template matching and predefined rules [4, 35], which were inflexible and required frequent manual updates. Machine learning approaches introduced more adaptability by learning query patterns from annotated datasets [42, 81], though they struggled with complex language constructs. Deep learning models, especially transformers like BERT [15] and GPT [11], enabled semantic understanding and laid the foundation for text-to-SQL systems [36]. Modern LLMs, such as PaLM [12] and GPT-4 [1], generate context-aware queries across multiple domains with minimal supervision. Retrieval-augmented generation (RAG) further enhances query precision by leveraging external knowledge [34]. Ongoing efforts focus on fine-tuning LLMs to improve accuracy and efficiency in converting natural language to structured queries [63, 64, 38].

## C  DETAILED METHODOLOGICAL PRELIMINARIES

**Data and Query Model:** We consider the dataset $\mathcal{D} = \{t_1, t_2, \ldots, t_n\}$ as a set of $n$ multi-modal tuples with no explicit values on specific attributes or metadata descriptions about them. This model complies with real-world needs, where personal devices such as cell phones have advanced cameras, enabling a vast volume of multi-modal data collection and sharing by any user of such devices. The

input query $\varphi$ is defined as a natural language description that corresponds to a subset of tuples $\mathcal{D}_\varphi \subseteq \mathcal{D}$, where $\mathcal{D}_\varphi = \{t'_1, \cdots, t'_k\}$ represents the target subset for retrieval.

**Nearest-Neighbor Search:** Nearest-neighbor search in traditional settings is a well-studied line of research where many algorithms and indices, such as Voronoi diagrams [31], tree data structures [76], and local sensitive hashing [24, 23] have been proposed for efficient exact and approximate query answering. In such settings, the data points are specified as points in a $d$-dimensional space, i.e., $t_i \in \mathbb{R}^d$, where the goal is to find the $k$ nearest neighbors to a query point $q \in \mathbb{R}^d$ based on a distance function $dist : \mathbb{R}^d \times \mathbb{R}^d \to \mathbb{R}$. That is, $k\text{-}\min_{t_i \in \mathcal{D}} dist(q, t_i)$.

Such approaches, however, are not directly applicable in our setting because of two main challenges. (a) Multi-modal tuples such as images are not represented as points in $\mathbb{R}^d$. and (b) The query is in the form of a (possibly complex) natural language (NL) phrase. Fortunately, vector representations (aka embeddings) [48, 29] have been proposed to address the first challenge. The state-of-the-art resolution to the second challenge is by utilizing the embeddings jointly trained on $\langle$tuple, text$\rangle$ pairs – which enables transforming the query and the tuples into the same embedding space and applying NN-search to answer the query. However, as we observe in § 5, while this idea works well for simple queries such as object detection, it fails for more complex queries.

**Vector Representations:** A vector representation $\mathcal{E} : dom(t) \to \mathbb{R}^{d_\mathcal{E}}$ is a transformation of the tuples to a high-dimensional vector of numeric values where the semantic similarity of tuples is proportional to the cosine similarity of their vector representations. Specifically, given a vector representation (aka *embedder*) $\mathcal{E}^\ell$, we present $\mathcal{E}^\ell(t_i)$ as $\vec{v}_i^\ell = \langle v_1, v_2, \cdots, v_{d_{\mathcal{E}\ell}} \rangle$. The distance between two vector embeddings $\vec{v}_i^\ell$ and $\vec{v}_j^\ell$ is computed based on their cosine similarity as

$$\delta_{i,j}^\ell = \delta(\vec{v}_i^\ell, \vec{v}_j^\ell) = 1 - \cos\left(\angle(\vec{v}_i^\ell, \vec{v}_j^\ell)\right) \tag{2}$$

Let $\Delta(t_i, t_j)$ be the (unknown) semantic distance of the tuples $t_i$ and $t_j$. A vector representation $\mathcal{E}^o$ is optimal if it captures the semantic similarity of tuples, i.e., $\delta_{i,j}^o \sim \Delta(t_i, t_j)$, $\forall t_i, t_j$. Formally,

$$\Delta(t_i, t_j) \geq \Delta(t_i, t_k) \quad \Leftrightarrow \quad \delta_{i,j}^o \geq \delta_{i,k}^o \tag{3}$$

In practice, however, vector representations are learned using a universe of available multimodal data. As a result, instead of guaranteeing Equation 3, the learned embeddings satisfy a (weaker) guarantee in expectation. That is, the *expected* distance between the embeddings of two tuples is proportional to their semantic distance, i.e., $\mathbb{E}\left[\delta_{i,j}^\ell\right] \sim \Delta(t_i, t_j)$, $\forall t_i, t_j$.

**Baseline Approach Limitations:** A vector representation that is jointly trained on multi-modal and textual data transforms text and images into the same vector-representation space. Therefore, given a natural language query, one can compute its embedding and find its nearest neighbor images. This approach, however, suffers from two major issues:

First, the paired texts used for training are usually simple sentences (e.g., [A photo of a dog]), unable to fully describe the rich information encoded in an image. Conversely, our goal is to answer complex queries describing a scene with multiple objects and their relationships (e.g., [A dog sitting by the fireplace, staring at the cat on the couch]). As a result, while such embeddings may perform reasonably on simple queries, their performance drops significantly on complex ones.

Second, as explained earlier, learned embeddings only preserve semantic similarities in expectation. As a result, relying solely on one vector representation for query answering may be misleading and inaccurate.

# D    PROOF OF THEOREM 1

**Theorem 1**    *For every tuple $t_i$ in a multi-modal dataset $\mathcal{D}$, let    $\delta_{\varphi,i}$, be the true distance of $t_i$ to $\varphi$.    Given a positive value $\gamma$,*

$$\Pr\left( \left( \frac{\bar{\delta}_{\varphi,i}}{\delta_{\varphi,i}} \geq (1+\gamma) \right) \vee \left( \frac{\bar{\delta}_{\varphi,i}}{\delta_{\varphi,i}} \leq (1-\gamma) \right) \right) \leq \mathbf{e}^{\frac{-m\,l\,\gamma^2\delta_{\varphi,i}}{3}} + \mathbf{e}^{\frac{-m\,l\,\gamma^2\delta_{\varphi,i}}{2}}$$

**Proof:** Let $\delta_{\varphi,i} = \delta\left(\vec{q}(\varphi), \vec{v}_i^o\right)$ be the true distance of $t_i$ to $\varphi$, $\forall t_i \in \mathcal{D}$. Replacing $\bar{\delta}_{\varphi,i}$ with the right-hand side of Equation 1, we get

$$\Pr\left(\frac{\bar{\delta}_{\varphi,i}}{\delta_{\varphi,i}} \geq (1+\gamma)\right) = \Pr\left(\frac{\frac{1}{m\,l}\sum_{j=1}^m \sum_{\ell=1}^l \delta\left(\mathcal{E}^\ell(\bar{\mathbf{g}}_j), \vec{v}_i^\ell\right)}{\delta_{\varphi,i}} \geq (1+\gamma)\right)$$

$$= \Pr\left(\sum_{j=1}^m \sum_{\ell=1}^l \delta\left(\mathcal{E}^\ell(\bar{\mathbf{g}}_j), \vec{v}_i^\ell\right) \geq (1+\gamma)m\,l\,\delta_{\varphi,i}\right)$$

Since $\mathbb{E}\left[\delta_{i,j}^\ell\right] = \delta_{i,j}^o$, $\forall t_i, t_j$ and $\mathbb{E}\left[\mathcal{E}^o(\bar{\mathbf{g}}_\varphi)\right] = \mathcal{E}^o(\mathbf{g}_\varphi)$,

$$\mathbb{E}\left[\delta\left(\mathcal{E}^\ell(\bar{\mathbf{g}}_j), \vec{v}_i^\ell\right)\right] = \delta_{\varphi,i}$$

As a result,

$$\mathbb{E}\left[\sum_{j=1}^m \sum_{\ell=1}^l \delta\left(\mathcal{E}^\ell(\bar{\mathbf{g}}_j), \vec{v}_i^\ell\right)\right] = m\,l\,\delta_{\varphi,i}$$

Next, applying Chernoff bound [46], we get

$$\Pr\left(\frac{\bar{\delta}_{\varphi,i}}{\delta_{\varphi,i}} \geq (1+\gamma)\right) \leq \mathbf{e}^{\frac{-m\,l\,\gamma^2\delta_{\varphi,i}}{3}}$$

Similarly,

$$\Pr\left(\frac{\bar{\delta}_{\varphi,i}}{\delta_{\varphi,i}} \leq (1-\gamma)\right) \leq \mathbf{e}^{\frac{-m\,l\,\gamma^2\delta_{\varphi,i}}{2}}$$

Therefore, applying the Union bound [46], we get

$$\Pr\left(\left(\frac{\bar{\delta}_{\varphi,i}}{\delta_{\varphi,i}} \geq (1+\gamma)\right) \vee \left(\frac{\bar{\delta}_{\varphi,i}}{\delta_{\varphi,i}} \leq (1-\gamma)\right)\right) \leq \mathbf{e}^{\frac{-m\,l\,\gamma^2\delta_{\varphi,i}}{3}} + \mathbf{e}^{\frac{-m\,l\,\gamma^2\delta_{\varphi,i}}{2}}$$

**Correction for Correlated Embedders:** The above proof assumes independence among the $l$ embedders. When embedders are correlated with an average pairwise correlation $\rho$, the effective sample size is reduced to $l_{\text{eff}} = \frac{l}{1+(l-1)\rho}$. In this case, the exponent in the bounds changes from $m \cdot l$ to $m \cdot l_{\text{eff}}$, yielding:

$$\Pr(\dots) \leq \mathbf{e}^{\frac{-m\,l_{\text{eff}}\,\gamma^2\delta_{\varphi,i}}{3}} + \mathbf{e}^{\frac{-m\,l_{\text{eff}}\,\gamma^2\delta_{\varphi,i}}{2}}$$

This adjustment reflects that correlated embedders provide less information gain than independent ones, resulting in a looser probability bound.

# E  DETAILED MECHANISMS OF THE INITIAL PROTOTYPE

## E.1  RANKINGS AGGREGATION AND EMBEDDER TRUST MECHANISM

To address the first challenge, NEEDLE employs a dynamic aggregation strategy that accounts for the varying reliability of its different embedders. Note that an embedder's performance is often not uniform across all tasks; for instance, one embedder might excel at distinguishing dog breeds but underperform in fruit classification. To handle this domain-specific performance, NEEDLE incorporates an embedder trust mechanism. This mechanism monitors each embedder's performance across various query topics, continuously adjusting topic-specific reliability scores to ensure the final aggregated result is intelligently weighted toward the most suitable embedders for the query at hand.

Consider a predefined set of topics $T$, where a natural language query $\varphi$ belongs to a specific topic $t = t(\varphi)^8$. Let $\mathbf{G} = \{\bar{\mathbf{g}}_1, \cdots, \bar{\mathbf{g}}_m\}$ be the set of guide images generated for $\varphi$, and $\mathbf{E} = \{\mathcal{E}^1, \cdots, \mathcal{E}^l\}$ be the set of embedders. For each guide image $\bar{\mathbf{g}}_j$, $R_j^i = \mathcal{R}^i(\varphi, \bar{\mathbf{g}}_j)$ represents the sorted list of $k$ nearest images in $\mathcal{D}$ to $\bar{\mathbf{g}}_j$ according to embedder $\mathcal{E}^i$. Let $R_j^i[r]$ be the $r$-th element of $R_j^i$.

For each topic $t \in T$, a weight $w_i^t \in [0, 1]$ is assigned to each embedder $\mathcal{E}^i$, indicating its reliability for queries within topic $t$. These weights are normalized such that $\sum_{i=1}^{l} w_i^t = 1$ for each topic $t$.

**Aggregation:** Given the ranked lists from all embedders and the query topic $t$, the objective is to aggregate these lists into a single final ranking $R_\varphi$ using the topic-specific reliability weights $w_i^t$. Let $I_h \in \mathcal{D}$ be a tuple returned by at least one embedder, and $rank(I_h, R_j^i)$ be its rank in list $R_j^i$. We adopt a monotonically decreasing position-importance function $\mathsf{S} : [k] \to [0, 1]$ from literature [13, 5, 19], specifically:

$$\mathsf{S}(i) = \begin{cases} \frac{1}{i} & i \leq k \\ 0 & \text{otherwise} \end{cases}$$

The aggregated score of $I_h$ is then computed as:

$$score(I_h) = \sum_{i : \mathcal{E}^i \in \mathbf{E}} w_i^t \cdot \mathsf{S}(rank(I_h, R_j^i)) \tag{4}$$

The top-$k$ tuples based on this score are returned as the final query output, $R_\varphi$.

**Dynamic Reliability-weight Adjustment:** To continually refine the embedder weights, NEEDLE incorporates a feedback mechanism. If, after presenting the top-$k$ results $R_\varphi$ to the user, a subset of results $\mathcal{T} = \{I_1', I_2', \ldots, I_s'\} \subseteq R_\varphi$ is marked as irrelevant[9], this feedback is used to update the reliability weights.

For the query's topic $t = t(\varphi)$, the partial loss of embedder $\mathcal{E}^i$ is computed as:

$$\Delta loss_\varphi(t, i) = \sum_{j : \bar{\mathbf{g}}_j \in \mathbf{G}} \sum_{I_h' \in \mathcal{T}} \mathsf{S}(rank(I_h', R_j^i)) \tag{5}$$

Only embedders that highly ranked the irrelevant results within topic $t$ are penalized. The topic-specific weights are then updated multiplicatively using the Multiplicative Weight Update Method (MWUM) [6]:

$$w_i^t(t + 1) = w_i^t(t) \cdot (1 - \eta \cdot \Delta loss_\varphi(t, i))$$

where $\eta \in (0, 1)$ is the learning rate. Finally, the weights for topic $t$ are normalized to ensure $\sum_{i=1}^{l} w_i^t = 1$.

### E.2 ANOMALY DETECTION IN QUERY IMAGES

The second challenge in our initial prototype involves ensuring the quality of the generated guide images. While foundation models are powerful, they can occasionally produce semantically off-target or poorly aligned images, which could degrade retrieval performance. To mitigate this, NEEDLE employs an outlier detection mechanism to identify and filter out potentially anomalous guide images based on their embeddings from multiple embedders. This process combines embeddings from all available embedders and uses a weighted approach to adjust for each embedder's reliability.

Given a set of $m$ guide images $\mathbf{G} = \{\bar{\mathbf{g}}_1, \bar{\mathbf{g}}_2, ..., \bar{\mathbf{g}}_m\}$ and $l$ embedders $\mathbf{E} = \{\mathcal{E}^1, \mathcal{E}^2, ..., \mathcal{E}^l\}$, each image $\bar{\mathbf{g}}_j$ yields $l$ embeddings: $\{\vec{v}_j^1, \vec{v}_j^2, \ldots, \vec{v}_j^l\}$, where $\vec{v}_j^i = \mathcal{E}^i(\bar{\mathbf{g}}_j)$. The goal is to detect anomalous guide images by analyzing these embeddings.

**Dimensionality Reduction using UMAP [45]:** Given the high-dimensional nature of the embeddings, we first apply dimensionality reduction using UMAP (Uniform Manifold Approximation and

---

[8]The topic $t$ of query $\varphi$ is identified using a topic classification mechanism or is part of the query input.

[9]We acknowledge that such feedback may not commonly be available after system deployment. However, crowdsourced feedback on a small set of topic-based benchmarks can be employed to calculate these weights before deploying a new embedder to our system.

Projection) to project them into a lower-dimensional space. For each embedder $\mathcal{E}^i$, the embeddings $\{\vec{v}_1^i, \vec{v}_2^i, \ldots, \vec{v}_m^i\}$ are reduced to a dimension $d$:

$$\vec{v'}_j^i = \text{UMAP}(\vec{v}_j^i) \in \mathbb{R}^d$$

where $d \ll \dim(\vec{v}_j^i)$.

**Outlier Detection using LOF [10]:** After dimensionality reduction, the Local Outlier Factor (LOF) algorithm is applied to identify outliers. LOF assigns an outlier score to each data point based on its neighborhood density. The LOF score for image $\bar{\mathbf{g}}_j$ with respect to embedder $\mathcal{E}^i$ is defined as:

$$\text{LOF}^i(\bar{\mathbf{g}}_j) = \text{LOF}(\vec{v'}_j^i)$$

Higher scores indicate a greater likelihood of $\bar{\mathbf{g}}_j$ being an outlier within the context of embeddings from $\mathcal{E}^i$.

**Aggregation of Outlier Scores:** The final outlier score for a guide image $\bar{\mathbf{g}}_j$ is computed by aggregating these individual scores, weighted by the reliability of each embedder:

$$S(\bar{\mathbf{g}}_j) = \sum_{i=1}^{l} w_i \cdot \text{LOF}^i(\bar{\mathbf{g}}_j)$$

**Anomaly Detection Threshold:** An image $\bar{\mathbf{g}}_j$ is flagged as an anomaly if its outlier score $S(\bar{\mathbf{g}}_j)$ exceeds a predefined threshold $\tau$:

$$S(\bar{\mathbf{g}}_j) > \tau \quad \Rightarrow \quad \bar{\mathbf{g}}_j \text{ is an anomaly}$$

The threshold $\tau$ is a hyperparameter that governs the system's sensitivity to anomalies. A higher $\tau$ promotes greater diversity in retrieved images, potentially increasing recall but risking lower precision, while a lower $\tau$ enforces stricter adherence to the query, leading to higher precision but potentially missing some relevant images. It is crucial to note that we tune $\tau$ specifically to target technical failures inherent to foundation models—such as solid black images triggered by safety filters or pure Gaussian noise—rather than penalizing stylistic diversity. This ensures that unique or creative guide images, which are valid representations of the query but might be statistically distant from the mode, are preserved to maintain the richness of the Monte Carlo sample. Our ablation study in Section 5.3 (Table 4) demonstrates the effectiveness of this mechanism.

# F  DETAILED EFFICIENCY MECHANISMS

## F.1  QUERY COMPLEXITY CLASSIFIER AND SHORT-CIRCUITING

To avoid image generation for simple queries, we use a **Query Complexity Classifier** to predict if a query is "simple" or "complex". A query is considered simple if existing methods (which we term "elementary methods", $\mathcal{M}$) are expected to return results with a high Average Precision (AP). Predicting complexity from linguistic features alone is often insufficient, so we use a lightweight, feature-based approach where features are derived from the elementary retrieval results themselves. These features form a vector $\vec{f}(\varphi) \in \mathbb{R}^D$ and include:

- **Mean Top-$K$ Cosine Similarity Scores:** For each elementary method $M_p$, we compute the average cosine similarity of its top-$K$ retrieved images, $R_{p,K}(\varphi)$. A higher score generally suggests a higher expected AP.

$$\bar{S}_{M_p,K}(\varphi) = \frac{1}{K} \sum_{I \in R_{p,K}(\varphi)} S_{M_p}(I, \varphi)$$

- **Top-$K$ Inter-Method Overlap Coefficients:** To quantify consensus, we compute the Jaccard index between the top-$K$ results of any pair of distinct methods $(M_p, M_q)$. High overlap is characteristic of simpler queries.

$$J(R_{p,K}(\varphi), R_{q,K}(\varphi)) = \frac{|R_{p,K}(\varphi) \cap R_{q,K}(\varphi)|}{|R_{p,K}(\varphi) \cup R_{q,K}(\varphi)|}$$

- **Confidence Deviation:** We measure the consistency of similarity scores using the standard deviation of the top-$K$ scores, $\sigma_{M_p, K}(\varphi)$. A lower standard deviation suggests higher confidence.

$$\sigma_{M_p, K}(\varphi) = \sqrt{\frac{1}{K-1} \sum_{I \in R_{p,K}(\varphi)} (S_{M_p}(I, \varphi) - \bar{S}_{M_p, K}(\varphi))^2}$$

This feature vector $\vec{f}(\varphi)$ is input to a pre-trained, lightweight regression model $\mathcal{C} : \mathbb{R}^D \rightarrow [0,1]$ (e.g., linear regression), which outputs a predicted AP score, $\text{AP}_{\text{pred}}(\varphi)$. If this score exceeds a threshold $\gamma_{AP}$, the query is classified as simple and short-circuited; otherwise, it is classified as complex and proceeds to the full pipeline.

### F.2 Implicit Metadata Generation

The implicit metadata generation module creates a positive feedback loop. After the full pipeline runs for a complex query, this module tags high-confidence image results with the query text. These new tags enrich our internal knowledge base, boosting the performance of the tag-based retrieval used by the Query Complexity Classifier.

The core challenge is to tag images with high confidence. We validate that the raw cosine similarity score from a multi-modal embedder serves as a reliable proxy for retrieval confidence. For a set of queries $\mathcal{Q} = \{\varphi_k\}_{k=1}^N$, we define the mean cosine similarity (MCS) for a query $q_k$ from an embedder $\ell$ as:

$$\text{MCS}_\ell(\varphi_k) = \frac{1}{|\mathcal{R}_{\varphi_k}|} \sum_{I \in \mathcal{R}_{\varphi_k}} \cos(\angle(\vec{v}_I^\ell, \vec{v}_{\varphi_k}^\ell)) \tag{6}$$

We then compute the Pearson correlation coefficient, $r$, between the MCS values and the corresponding true Average Precision (AP) values across all queries:

$$r = \frac{\sum_{k=1}^N (\text{MCS}_\ell(\varphi_k) - \overline{\text{MCS}})(\text{AP}(\varphi_k) - \overline{\text{AP}})}{\sqrt{\sum_{k=1}^N (\text{MCS}_\ell(\varphi_k) - \overline{\text{MCS}})^2 \sum_{k=1}^N (\text{AP}(\varphi_k) - \overline{\text{AP}})^2}} \tag{7}$$

Our empirical validation (Figure 8) shows a strong, direct correlation. Based on this, our tagging strategy is as follows: after a retrieval, we assign the query $\varphi$ as a tag to any retrieved image $I$ where the similarity score $\cos(\angle(\vec{v}_I^\ell, \vec{v}_\varphi^\ell))$ exceeds a fine-tuned confidence threshold $\tau$. This process continually improves our metadata, leading to more short-circuiting and higher efficiency over time.

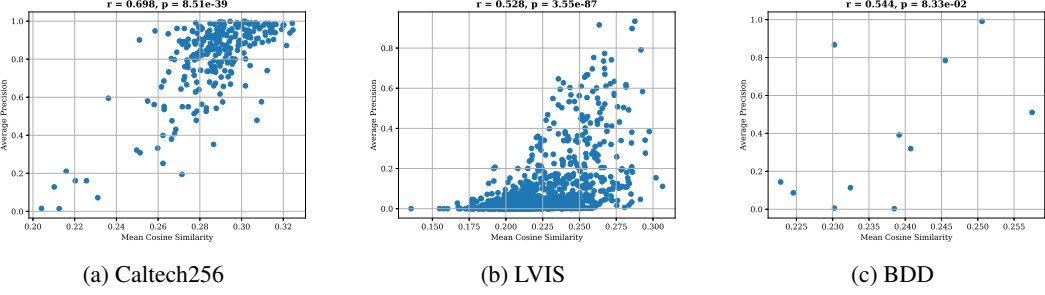

| (a) Caltech256 | (b) LVIS | (c) BDD |

Figure 8: Correlation between CLIP cosine similarity and Mean Average Precision (MAP) across various benchmarks. The Pearson correlation coefficient ($r$) and its associated p-value are displayed within each subplot.

## G System Development

### G.1 Development Objectives

As one of our primary objectives, we aim to develop NEEDLE as an open-source, deployment-ready application that both researchers and developers can easily adopt. Specifically, we considered a

set of design objectives, outlined in the following; later, in § G.2, we describe how our system implementation fulfills these objectives.

**Database Functionality and CLI Integration:** NEEDLE should be easily installable across various operating systems and offer an intuitive command-line interface (CLI) for efficient user interaction.

**Deployment Readiness and Versatility:** NEEDLE should be engineered for immediate deployment, operating out-of-the-box on a wide range of system configurations, including resource-limited personal computers.

**Configurability and Abstract Database Modes:** Users should have the freedom to select from multiple operational modes (fast, balanced, accurate) that balance retrieval accuracy and latency according to their specific application needs. Additionally, NEEDLE should allow for customization beyond these predefined modes.

**GPU Acceleration and Scalability:** Given the computational intensity of image indexing, NEEDLE should automatically leverage available GPU resources and support workload distribution across multiple GPUs to ensure scalable performance.

**Modular Embedder Architecture:** As embedders are central to NEEDLE's functionality, the architecture should facilitate easy integration and upgrading of new embedder models, ensuring the system remains at the forefront of technological advancements.

**Multi-Directory Management:** NEEDLE should be capable of managing multiple image directories concurrently. It should allow users to add, remove, or toggle directories for search while indexing new directories in the background without interrupting ongoing queries.

**Robustness and Consistency Maintenance:** NEEDLE should perform consistency checks upon restart after downtime to synchronize any database changes. It should continuously monitor designated directories for modifications, ensuring the database index remains aligned with the file system.

**Integration with External Image Generators:** To facilitate the generation of guide images, NEEDLE should provide connection wrappers for leading proprietary and open-source image generation services (e.g., DALL-E, Google Imagen, Replicate). Furthermore, it should allow users to integrate any image generator that complies with the specified request schema.

**Flexible Output Formats and Enhanced Usability:** Recognizing that NEEDLE may serve as a component within larger pipelines, it should support multiple output formats, including JSON, YAML, and human-readable text. Additionally, built-in query previews should be provided to improve the overall user experience.

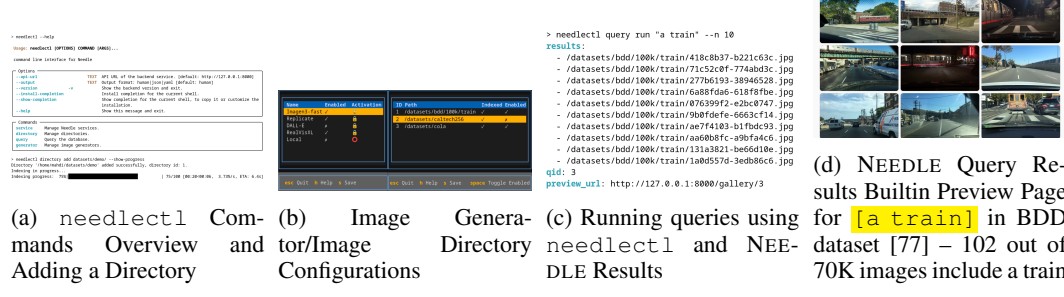

(a) `needlectl` Commands Overview and Adding a Directory  (b) Image Generator/Image Directory Configurations  (c) Running queries using `needlectl` and NEEDLE Results  (d) NEEDLE Query Results Builtin Preview Page for `[a train]` in BDD dataset [77] – 102 out of 70K images include a train

Figure 9: Screenshots of key interfaces and features of NEEDLE.

## G.2 SYSTEM DETAILS

Having defined our development objectives, we can now delve into the NEEDLE's implementation details.

**Architecture:** NEEDLE by nature requires multiple services to work together. It needs a service for handling Image Generation, a service for handling *Directory Integration* and *Indexing Progress*, and a service for storing *calculated embeddings* and *kNN lookup*. In order to handle these multiple

sources of responsibilities, we have designed NEEDLE with a microservice architecture, delegating correlated responsibilities to their respective services. NEEDLE's services are:

**Backend:** This microservice serves as the gateway to the core functionalities of NEEDLE. It encapsulates the primary operations of the system, exposing APIs such as `/search`, `/directory`, and `/query`. Additionally, it manages the initialization of embedders and oversees the image indexing process.

**Database:** The backend container needs persistent storage to store directory structures, indexing progress, and other critical metadata. NEEDLE is agnostic to the choice of the database, but in the current implementation, we use `PostgreSQL`[10] for this goal.

**Vector Store:** NEEDLE requires a vector store that supports embedding, indexing, and inner product searches. This service is used to store the outputs of various embedders and to efficiently execute k-nearest neighbor (kNN) searches for a given query. In the current implementation, we have used `Milvus`[11] for this purpose.

**Image Generator Hub:** This service manages the image generation process by providing wrappers for renowned services such as DALL-E, Imagen, and Replicate. It also supports custom image generation models that conform to its defined schema. Moreover, the hub offers fallback mechanisms and priority settings to maximize robustness and ensure reliable image generation.

**Command-Line Interface:** NEEDLE requires an intuitive interface to work with the backend service and apply the user's requests. This service is the exposed UI of NEEDLE to the end user.

Since the individual services have their own dependencies and requirements, we containerized all core services using `docker` to mitigate dependency conflicts and make deployment easier across various operating systems. We employ `docker compose` to orchestrate these containers, establish a shared network, and configure the individual services. Additionally, to eliminate the need for direct Docker manipulation, we developed a Command-Line Interface (CLI), `needlectl`, as a standalone binary. `needlectl` installs into the user's path and provides seamless access to the NEEDLE core.

**Technical Stack and Frameworks:** Backend and ImageGeneratorHub services built using `FastAPI` framework[12]. For GPU-intensive tasks, `PyTorch`[13] is leveraged to distribute workloads across available GPUs, ensuring efficient processing.

Embedder management is streamlined via integrations with Hugging Face and the `timm` library[14], facilitating the initiation and upgrading of embedding models. The CLI client (`needlectl`) is developed using the `Typer`[15] library to build intuitive command-line applications, while `textual`[16] is utilized to manage terminal user interfaces (TUIs).

Standalone Python libraries are packaged with `PyInstaller`[17], simplifying the distribution and execution. Installation, upgrading, and uninstallation processes are automated using `Bash` scripts, ensuring a smooth user experience. Containerization and orchestration are achieved with `Docker`[18] and `Docker Compose`, respectively.

For documentation, we employ `mdBook`[19] to create comprehensive and well-structured guides. Finally, all of the build, development, and versioning processes are automated via CI/CD pipelines using GitHub Actions[20].

---

[10]`https://www.postgresql.org/`
[11]`https://milvus.io/`
[12]`https://fastapi.tiangolo.com/`
[13]`https://pytorch.org/`
[14]`https://github.com/huggingface/pytorch-image-models`
[15]`https://typer.tiangolo.com/`
[16]`https://textual.textualize.io/`
[17]`https://pyinstaller.org/en/stable/`
[18]`https://www.docker.com/`
[19]`https://rust-lang.github.io/mdBook/`
[20]`https://github.com/features/actions`

Table 6: Embedder Models and Their Weights

| Architecture | Model Name | Weight |
|---|---|---|
| eva[18] | eva02_large_patch14_448 | 0.8497 |
| regnet[73] | regnety_1280 | 0.8235 |
| dinoV2[49] | vit_large_patch14_reg4_dinov2 | 0.8235 |
| CLIP[51] | vit_large_patch14_clip_336 | 0.8146 |
| convnextV2[72] | convnextv2_large | 0.8184 |
| bevit[8] | beitv2_large_patch16_224 | 0.7660 |

**Command-Line Interface and APIs:** NEEDLE offers a robust CLI that enables seamless interaction with its core services. The needlectl is provided as a standalone binary, allowing users to execute commands directly without requiring additional dependency installations. This design ensures that non-expert users can easily operate NEEDLE. The CLI commands and options have been carefully designed to be intuitive and straightforward. The command structure adheres to the following format:

```
needlectl [--global options] [component] [action] [--action options]
```

This structure supports clear and consistent command usage with the following key components (some illustrative examples are provided in Listing 10):

- service: for managing NEEDLE-related tasks

- directory: for handling image directory operations

- query: for managing active queries

- generator: for interfacing with image-generation functionalities

```
needlectl service start      # Starts the Needle services
needlectl service log        # Retrieves service logs
needlectl directory add /path/to/image/dir --show-progress
# Adds an image directory for indexing and monitoring,
# displaying progress bar
needlectl --output json query run "a wolf"
    --num-engines 2
    --num-images-to-generate 4
    --image-quality LOW
# Executes a query using two generators,
# each generating four low-quality images, with JSON output
```

Figure 10: Illustration of needlectl commands

**Installation and Deployment:**

NEEDLE is installed using a single one-liner command that executes a Bash script. This script automatically verifies system prerequisites and checks for GPU availability. It detects the current operating system and downloads the appropriate configuration. During installation, users are prompted to select a preferred database mode—Fast, Balanced, or Accurate—which optimizes various parameters such as the number of embedders, default image resolution for generation, default number of generators, number of images generated per query, and HNSW[43] index construction parameters (e.g., M and ef). Finally, the installation script guides users through the initial steps to start and utilize the NEEDLE service.

# H    ADDITIONAL EXPERIMENTS

## H.1    DETAILS OF BASELINE MODELS

**1.    CLIP [51]:**  Developed by OpenAI[21], this model uses a ViT-B/32 image encoder and a Transformer-based text encoder.  It learns a shared embedding space via contrastive learning on large-scale image–text pairs, and its widespread use and benchmarking make it a standard reference. We use CLIP as our standard baseline and designate categories where CLIP's average precision (AP) is below 0.5 as *"hard categories"*—instances where CLIP struggles to produce satisfactory results. This filtered subset is then used to demonstrate the enhanced capability of alternative baselines in effectively handling rare or difficult queries.

**2.    ALIGN [25]:**  Originally developed by Google, ALIGN (A Large-scale ImaGe and Noisy-text Embedding) employs a dual-encoder architecture similar to CLIP, using separate encoders for images and text trained with a contrastive loss.  However, while CLIP (in our `clip-vit-base-patch32` variant) uses a Vision Transformer for image encoding, ALIGN typically uses a CNN image encoder along with a Transformer for text. Moreover, the original ALIGN was trained on a much larger, noisier dataset (over 1.8 billion image–text pairs) than CLIP. In our experiments, we use an open-source version[22] that preserves the core architecture but is trained on different data, achieving even better accuracy on some benchmarks.

**3. FLAVA [58]:**[23] a unified multimodal model developed by Facebook that jointly learns representations for images and text.  Unlike CLIP, which primarily relies on a contrastive approach applied to image-text pairs, and ALIGN, which emphasizes scaling up representation learning using noisy text supervision with separate encoders, FLAVA adopts a holistic pre-training strategy. It integrates both unimodal and multimodal objectives, enabling FLAVA to capture richer semantic interactions across modalities.

**4.    BLIP + MiniLM:** This pipeline approach first converts images into descriptive captions using BLIP [37][24]—a state-of-the-art image captioning model from Salesforce known for its high-quality, informative captions.  The generated captions are then transformed into embeddings with MiniLM [69][25], a robust text encoder widely adopted in industrial applications.  This decoupled strategy not only leverages mature text retrieval systems but also enables independent optimization of the captioning and text-embedding stages. As a result, it has become a strong baseline and is employed in many commercial products, offering enhanced interpretability and scalability compared to end-to-end models like CLIP.

**5.    CoCa [78]:** CoCa (Contrastive Captioners) is an image-text foundation model that unifies contrastive learning and generative captioning within a single encoder-decoder architecture.  Unlike dual-encoder models that are trained only with a contrastive loss, CoCa's image encoder is trained with both a contrastive loss (matching images to text) and a captioning loss (generating descriptive text).  This dual-objective pre-training enables CoCa to excel at a wide range of downstream tasks, including zero-shot image retrieval, image captioning, and visual question answering, making it a powerful and versatile baseline.

**6. PlugIR [33]:** PlugIR (Interactive Text-to-Image Retrieval) is a framework that enhances existing retrieval models by leveraging the capabilities of a Large Language Model (LLM). Rather than being a standalone retrieval model, PlugIR uses an LLM as an intelligent query-rewriting module. For a given input text query, the LLM generates multiple, diverse, and descriptive reformulations of that query.  These expanded queries are then embedded using a base vision-language model (such as CLIP), and their results are aggregated to form the final ranked list. This "plug-and-play" approach

---

[21]https://huggingface.co/openai/clip-vit-base-patch32

[22]https://huggingface.co/kakaobrain/align-base

[23]https://huggingface.co/facebook/flava-full

[24]https://huggingface.co/Salesforce/blip-image-captioning-base

[25]https://huggingface.co/sentence-transformers/all-MiniLM-L6-v2

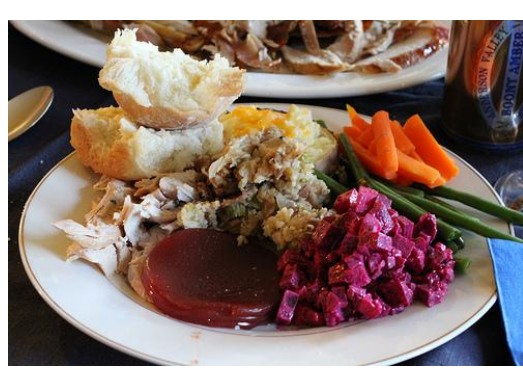 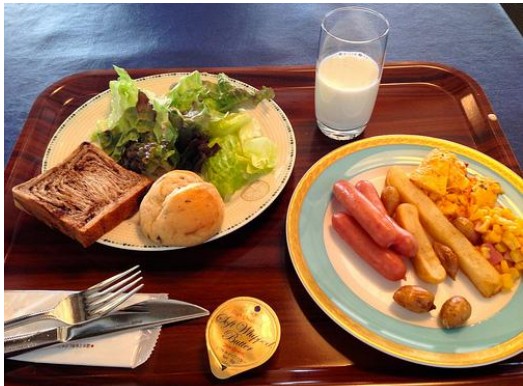

(a) Blue napkin to the right of white plate  (b) White napkin to the left of blue plate

Figure 11: Illustration of COLA compositional benchmark queries, using each image description, where the goal is to identify the correct image

allows for significant retrieval improvements by combining the reasoning power of an LLM with the strong visual-semantic alignment of existing models, all without requiring any re-training.

**8. E5-V [27]:** E5-V is a state-of-the-art multimodal embedding model that adapts a Multimodal Large Language Model (MLLM) to generate universal text and image embeddings. By fine-tuning the model to align textual and visual inputs into a single dense vector space, E5-V leverages the reasoning capabilities of large language models to capture nuanced semantic relationships often missed by traditional dual-encoder models.

## H.2 DATASET DETAILS

**Object Detection Datasets:**

1. **Caltech256 [21]**: Contains 30,607 images spanning 256 object categories. Its diverse object classes make it a useful benchmark for assessing the robustness of detection and retrieval models.

2. **MS COCO [39]**: With over 118K images and 80 object categories, MS COCO is widely used for object detection and segmentation. Its comprehensive annotations and varied scene compositions provide a challenging testbed.

3. **LVIS [22]**: offers instance segmentation with a long-tail distribution of over 1,200 categories. Its focus on rare and fine-grained objects is critical for evaluating retrieval performance on less frequent classes.

4. **BDD100k [77]**: This dataset is tailored to urban driving scenarios with detailed annotations, making it valuable for testing retrieval in real-world, dynamic contexts.

**Natural Language Query Datasets:**

For evaluating the performance of various baselines on complex natural language queries, we focus on scenarios where models must differentiate between two highly similar images that exhibit nuanced differences. In these settings, the model is expected to identify the correct image by carefully attending to subtle details in both the visual content and the corresponding query. To rigorously assess these capabilities, we utilize several compositional benchmarks:

5. **COLA [54]**: As a benchmark for compositional text-to-image retrieval, this dataset features nuanced captions that require the system to distinguish between multiple similar images by capturing subtle semantic and spatial details.

6. **Winoground [61]**: Designed to evaluate visio-linguistic alignment, Winoground presents pairs of images and captions that differ only in their compositional structure. This dataset challenges models to accurately map nuanced language to the corresponding image, serving as a stringent test of fine-grained retrieval performance.

7. **NoCaps [3]**: A large-scale benchmark for novel object captioning, featuring images from Open Images with human-generated captions. It is crucial for evaluating a model's ability to handle zero-shot retrieval on images containing objects not commonly found in standard captioning datasets.

8. **SentiCap [44]**: Derived from datasets like MS COCO, SentiCap provides images with captions specifically annotated for positive or negative sentiment. It allows for evaluating retrieval systems' ability to understand and retrieve images based on subjective or emotional language cues, moving beyond purely factual descriptions.

### H.3 Evaluation Metrics

We use different evaluation metrics for object detection and complex natural language queries benchmarks. In object detection benchmarks, we define each benchmark dataset $\mathcal{D}$ as a set of images $\mathcal{D} = \{t_1, t_2, \ldots, t_p\}$, where each image $t_i$ comprises a set of objects $t_i = \{o_1, o_2, \ldots, o_s\}$, with $o_j$ representing an individual object within image $t_i$. For each benchmark, we compute the union of all objects present across all images, denoted as $\bigcup_{i=1}^{n} t_i = \bigcup_{i=1}^{n} \{o_1, o_2, \ldots, o_s\}$, and utilize each unique object $o_j$ within this union as a query $q$ for each retrieval engine. For a given query $q$, we retrieve a ranked list of files $\mathcal{F} = \{f_1, f_2, \ldots, f_n\}$ ($n = 60$), representing the retrieved images. We construct a list of relevance scores $\mathcal{R} = \{r_1, r_2, \ldots, r_n\}$, where $r_i = 1$ if $q$ is present in image $f_i$, and $r_i = 0$ otherwise. We start list indexing from 1, so by definition $\mathcal{R}[j] = r_j$. We assume access to a function $C(.)$, which gets an object as input and returns the number of images that this object appears in. We are interested in retrieving at most 10 positive instances. Therefore, we define the effective number of positives as: $ep = \min(10, C(q))$.

Given $\mathcal{R}$ and $ep$, We assess the performance of the retrieval using the following metrics:

**Mean Recall at $k$ (R@k):** $r@k = \frac{\sum_{i=1}^{k} \mathcal{R}[i]}{ep}$. A higher r@k indicates that a larger proportion of the relevant images (up to 10) are retrieved in the top $k$ positions, we report mean r@k (R@k) which is the average of r@k over all objects in the dataset.

**Precision at $k$ (P@k):** $p@k = \frac{\sum_{i=1}^{k} \mathcal{R}[i]}{k}$. A higher p@k reflects that a greater proportion of the retrieved images are relevant, which is crucial when the user examines only the top results, same as before, we report the averaged out P@k over all object in the dataset.

**Mean Average Precision (MAP):** measures the average precision (AP) over all queries.

$$AP = \frac{1}{ep} \sum_{i=1}^{n} \left( \frac{\sum_{j=1}^{i} \mathcal{R}[j]}{i} \right) \cdot r_i,$$

where $r_i = \mathcal{R}[i]$ and the effective number of positives is defined as $ep = \min(10, C(q))$. The overall mean average precision (MAP) is then the average of $AP$ over all objects in the dataset. MAP provides a single-figure measure that considers both the precision and recall across the entire ranked list, rewarding systems that rank relevant images higher.

**Mean Reciprocal Rank (MRR):** computes the reciprocal rank for each query and then averages these values over all queries. For a given query $q$, let $k_q$ be the smallest index such that $\mathcal{R}[k_q] = 1$, indicating the rank of the first relevant image. The reciprocal rank for $q$ is defined as:

$$RR(q) = \begin{cases} \frac{1}{k_q}, & \text{if } \exists\, k \text{ with } \mathcal{R}[k] = 1, \\ 0, & \text{otherwise.} \end{cases}$$

The Mean Reciprocal Rank (MRR) is the average value over all queries. MRR emphasizes the importance of retrieving at least one relevant image as early as possible for ranking.

For the complex natural language query experiments, we evaluate our method using both MRR and Pairing Accuracy. In these experiments, the baseline is given two images, $t_1$ and $t_2$, along with two captions, $c_1$ and $c_2$, and each caption must be assigned to its corresponding image correctly. Pairing Accuracy is defined as the number of correct assignments divided by the total number of queries. Note that the pairing accuracy for a random baseline is 0.25% (chance of choosing the correct combination over all possible combinations).

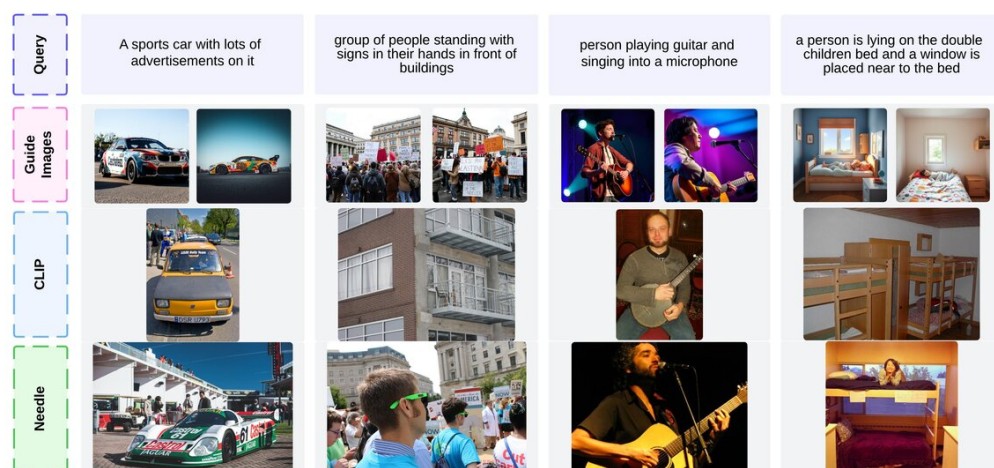

Figure 12: Illustration of Complex Natural Language Queries extracted from NoCaps [3], Guide images, and CLIP vs. NEEDLE results.

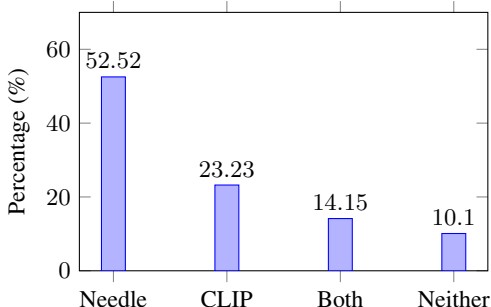

Figure 13: Results from the human evaluation case study on the LVIS dataset. Among the queries with a clear preference, NEEDLE was chosen in approximately 70% of cases, underscoring its practical effectiveness in retrieving relevant images.

### H.4 VISUAL EXAMPLES OF COMPLEX QUERIES

Figure 12 provides visual examples of how NEEDLE handles complex natural language queries compared to CLIP. The examples demonstrate the effectiveness of using synthetic guide images for retrieval tasks that require understanding of complex scene descriptions and object relationships.

### H.5 CASE STUDY

To further validate NEEDLE's practical applicability, we conducted a case study with human evaluators. We utilized the LVIS dataset as our primary image repository, as it contains over 100k images spanning various categories, ensuring that any randomized query would likely have related images. This made LVIS an ideal choice for our case study.

Each human evaluator was asked to write five queries of their choice, with varying levels of complexity. For each query, we retrieved 20 relevant images from both CLIP and NEEDLE, displaying the results side by side with randomized orientations for each query. Evaluators were then asked to select the more relevant results, choosing from four options: "Left is better," "Right is better," "Both are good enough," and "Neither is good enough". Then, we recorded the engine behind that option as the preferred engine.

A total of 20 human evaluators participated in the study, contributing 100 queries. The queries ranged from simple ones like "Celery stew" to more complex descriptions such as "A person walking on the sidewalk of a river in Fall at Chicago.", Figure 13 presents the results of this user study. If we consider only the queries where one engine is preferred over the other, NEEDLE is chosen as the better engine in almost 70% of cases.

To provide further insight into the diversity of the evaluation, Table 7 lists additional representative queries collected during the user study, categorized by their varying levels of descriptive complexity.

Table 7: Representative user queries from the case study ranging from simple objects to complex scenes.

| User Query |
| --- |
| Celery stew |
| A red Subaru parked next to a white Camry |
| A PhD student defending his dissertation |
| Presidential debates last night |
| Two cats and a dog playing together |
| Old man playing chess in the park with his friends |

## THE USE OF LARGE LANGUAGE MODELS (LLMS)

In the spirit of transparency and to reflect modern research practices, we acknowledge the use of Large Language Models (LLMs) as assistive tools in the preparation of this manuscript and the development of the associated software.

Throughout the writing process, we utilized LLMs to refine prose, improve clarity, and correct grammatical errors. The models served as an advanced editing tool to help articulate our ideas more effectively. It is important to state, however, that all core concepts, methodological designs, experimental analyses, and conclusions presented in this paper are the original contributions of the authors.

Similarly, during the development of the NEEDLE, LLMs were employed as a debugging and code-completion aid. They were particularly useful for identifying issues in code segments, suggesting solutions to programming errors, and accelerating the implementation of standard software components. The core architecture of NEEDLE and its novel algorithms were designed and implemented entirely by the authors. We believe that acknowledging the role of these powerful tools is an important step in maintaining the integrity and transparency of the modern research process.

