# OpenReview forum: "Needle: A Generative AI-Powered Multi-modal Database for Answering Complex Natural Language Queries"
_ICLR.cc/2026/Conference — Submitted to ICLR 2026_

### Official Review · Reviewer_aWQV · 2025-10-31

**Soundness:** 3
**Presentation:** 3
**Contribution:** 3
**Rating:** 6
**Confidence:** 3

**Summary:**

This paper introduces NEEDLE, a novel system for complex natural language query answering in multi-modal databases, specifically focusing on image retrieval. The core innovation is a Generative AI-powered Monte Carlo method that reframes the text-to-image search problem as an image-to-image search. Instead of relying on traditional contrastive learning, NEEDLE uses foundation models to generate synthetic "guide" images from the text query. These guide images, acting as stochastic representations of the query's intent, are then used with an ensemble of image embedders to perform nearest-neighbor search against the database. The system is enhanced with several optimizations, including a query complexity classifier to bypass expensive generation for simple queries and an implicit metadata generation loop for continuous improvement.

**Strengths:**

1. The proposed method of using generative models to create synthetic guide images is a novel and promising approach to bridge the semantic gap between complex text queries and image data. It cleverly transforms a cross-modal retrieval task into a more direct image-to-image search problem.

2. The system design is thorough and practical. Beyond the core theoretical contribution, the authors have developed a complete, open-source system with crucial efficiency optimizations like the Query Complexity Classifier and an Implicit Metadata Generation mechanism, which address the practical latency concerns of on-the-fly image generation.

3. The ablation studies are extensive and insightful. They effectively dissect the system's performance, analyzing the impact of key hyperparameters such as the number of guide images, the number of embedders, the choice of foundation models, and the effect of outlier detection, which validates the design choices.

**Weaknesses:**

1. The primary performance bottleneck and a significant practical challenge is the latency and computational cost associated with on-the-fly image generation by large foundation models. While the paper proposes optimizations to mitigate this, the reliance on this step for complex queries may still hinder real-time performance in large-scale, interactive applications.

2. The effectiveness of the system is heavily dependent on the quality of the generative foundation models and the image embedders. Any inherent biases, limitations, or failures of these underlying models (e.g., difficulty with compositional ordering, as noted by the authors) will be directly inherited by NEEDLE, potentially impacting its reliability and fairness.

3. The system's complexity is quite high, involving multiple components (generative models, multiple embedders, a vector store, a query classifier, etc.). This could present challenges for deployment, maintenance, and reproducibility compared to simpler, end-to-end contrastive learning models.

**Questions:**

N/A

---

> ### Author Response · Authors · 2025-11-21
>
> We thank the reviewer for their positive assessment and for recognizing the novelty of our generative approach in bridging the semantic gap for complex queries. We address your concerns regarding latency, model dependency, and complexity below.
>
> *W1. "The primary performance bottleneck... is the latency... associated with on-the-fly image generation... may still hinder real-time performance."*
>
> **Response:** We agree that image generation is computationally intensive. However, we view this as a necessary trade-off to achieve retrieval accuracy on complex queries where standard models fail. To make this practical for real-time applications, we designed the Query Complexity Classifier (Section 4.1) to strictly limit this latency cost to only the "complex" queries that actually require it; simple queries are routed to fast, traditional paths. Furthermore, the system is designed to become faster over time. The Implicit Metadata Generation mechanism creates a feedback loop: as complex queries are processed and results are verified, we tag the images with the query text. This converts future instances of similar queries into simple metadata lookups, progressively reducing the system's reliance on the expensive generation pipeline.
> Finally, we would like to clarify that our goal was not to *"beat"* standard baselines on runtime, but to prove that for a reasonable and **acceptable** production-level efficiency cost, we can deliver superior retrieval accuracy, as demonstrated in our experimental evaluation. We believe this trade-off makes Needle a powerful and practical solution. To demonstrate this viability, we have released Needle as a fully open-source, production-ready system, containerized for deployment on platforms like Linux and Mac.
>
> *W2. "The effectiveness of the system is heavily dependent on the quality of the generative foundation models... inherent biases... will be directly inherited."*
>
> **Response:** This is a valid point, as "garbage in, garbage out" is a risk with any generative system. However, Needle's architecture is specifically designed to mitigate the failures of individual models. By utilizing a Monte Carlo framework (averaging over m guide images) and an ensemble of embedders, we smooth out the variance and specific biases of any single model. Additionally, we employ the Embedder Trust Mechanism and Outlier Detection (LOF) to actively filter out low-quality or anomalous generations before they impact the retrieval results. Finally, Needle is designed to be model-agnostic ; as foundation models improve in their ability to handle compositionality, Needle’s performance will naturally improve without requiring architectural changes.
>
> *W3. "The system's complexity is quite high... could present challenges for deployment, maintenance, and reproducibility."*
>
> **Response:** We acknowledge that Needle is more complex than a single end-to-end model. To address this, we invested significant effort into making the system "production-ready" via a modular microservice architecture. We containerized all components using Docker and developed a standalone CLI (needlectl) to abstract this complexity away from the user. The entire system **can be installed and deployed with a single script**, ensuring that while the internal logic is sophisticated, the user experience and reproducibility remain straightforward.

---

### Official Review · Reviewer_yRLh · 2025-11-01

**Soundness:** 3
**Presentation:** 3
**Contribution:** 2
**Rating:** 4
**Confidence:** 4

**Summary:**

This paper propose *Needle*, a retrieval system for image databases that answers complex natural-language queries by using a text-to-image generator to create several "guide" images for the query. The authors propose to embed those guides with an ensemble of image embedders, by running multiple k-NN lookups over the target image collection, and aggregating the rankings (with optional per-topic "trust weights"). The core idea is framed as a Monte Carlo estimator of the (unknown) query-image distance: average distances between many guide-image embeddings and candidate images to approximate the true semantic distance. The authors report their results on CLIP, ALIGN, FLAVA and a BLIP+MiniLM pipeline on object-centric datasets and compositional retrieval benchmarks (COLA, Winoground, NoCaps, SentiCap).

**Strengths:**

- The proposed method is simple yet effective: the authors turn a complex text-to-image retrieval problem into many image-to-image searches with an engineering and system architecture focus (vector store, caching, outlier filtering, embedder weighting)

- The proposed system is modular: any generator and any embedder can be swapped (with weighting embedders and filtering out bad guides)

- Strong performance on R@10/MAP/MRR and pairing accuracy against the included baselines.

**Weaknesses:**

- The paper claims to be the first to use generative models to synthesize query-side signals for multimodal retrieval, but prior work has already explored generative modeling for image retrieval [1, 2, 3].

- In section 2, theorem 1 is based on the assumption that "Each guide tuple is considered an i.i.d. sample from a distribution whose mean is the ideal (but unknown) tuple that perfectly represents the query." However, this relies heavily on each guide tuple to be unbiased. There is no experiment estimating the bias/variance of the estimator as a function of number of guides and embedders, nor any stress test when generators systematically miss relations, which is particularly common for compositional prompts.

- This paper omits several strong modern baselines widely used for retrieval and compositional reasoning, e.g., OpenCLIP ViT-bigG/EVA-CLIP, SigLIP families, or recent compositional adapters for CLIP/FLAVA trained specifically on attribute-object binding (which COLA’s authors show matter).

- In the evaluations, the hard set definition uses CLIP AP < 0.5 to decide difficulty and then reports big gains over CLIP on that subset.

- Consequently, having the Related Work in the supplementary material is a big downside -- given the ICLR reviewer guidelines (It is not necessary to read supplementary material (...)), this seems a major flaw in this paper. I strongly recommend moving this section to the main paper for future resubmissions; this helps reviewers and readers understand the position of the paper, its claims and differences with prior and concurrent work.

[1] Zhang, Yidan, et al. "Irgen: Generative modeling for image retrieval." European Conference on Computer Vision. Cham: Springer Nature Switzerland, 2024.

[2] Li et al. Generative Cross-Modal Retrieval: Memorizing Images in Multimodal Language Models for Retrieval and Beyond (ACL 2024)

[3] Kim, Sungyeon, et al. "GENIUS: A generative framework for universal multimodal search." Proceedings of the Computer Vision and Pattern Recognition Conference. 2025.

**Questions:**

- When generators systematically miss relations (e.g., "a lightbulb surrounding some plants"), how biased is the distance δ between the query φ and any tuple t?

- Why not including SigLIP, OpenCLIP bigG/EVA-CLIP, and COLA-adapted multimodal layers?

- How sensitive are results to prompt phrasing and seed selection for the guide images?

This is more of a suggestion: SIGIR or CIKM seems like a better fit for this submission.

---

> ### Author Response · Authors · 2025-11-21
>
> We thank the reviewer for their detailed feedback. We address your specific concerns regarding novelty, theoretical assumptions, and baselines below.
>
> *W1. "The paper claims to be the first... but prior work has already explored generative modeling... [1, 2, 3]."*
>
> **Response:** We thank the reviewer for these references. This is a very active area of research, and we agree these are important comparisons. As noted in our responses to other reviewers, an earlier version of our paper was made publicly available **prior** to the publication of the works cited. While we cannot link to this pre-print here to maintain anonymity, this timing establishes that these recent papers are either concurrent with or subsequent to our work.
>
> We acknowledge that this is a rapidly evolving field. In light of this concurrent literature, we will add a **remarks section** and **refine our novelty statement**. We will update our claim to be more precise, clearly distinguishing Needle's specific contributions, such as our Generative-based Monte Carlo framework and practical optimizations in system design, from these related approaches.
>
> *W2. Q1."Theorem 1 is based on the assumption that 'Each guide tuple is considered an i.i.d. sample'... However, this relies heavily on each guide tuple to be unbiased... When generators systematically miss relations... how biased is the distance?"*
>
> **Response:** We agree with the reviewer that Theorem 1 relies on each guide tuple to be unbiased. We explicitly acknowledge this in Section 5.2 that current generators often fail on complex compositional prompts.
> Practically, however, our Monte-carlo based framework can reduce the impact of individual models by generating samples using diverse set of foundation models. Ideally by generating m samples using m different models the effect of the inherence biases of individual models are minimized. Moreover, one can learn the bias of the models over time, and penalize the samples generated by them by taking a weighted average, where the biased models have smaller weights.
> We have added this to the revised manuscript.
>
> The very purpose of our Generative-based Monte Carlo framework is to mitigate these practical failures. By sampling multiple guide images ($m$) and using multiple embedders ($l$), our framework averages out the stochasticity, biases, and specific failures of any single generator or embedder.
>
> We have experiments that shows :
> - increasing m and l (guide images and embedders) consistently improves MAP (Figure 4).
> - Ensembling multiple foundation models also improves MAP (Figure 6a).
> - A full ensemble outperforms any single embedder.
>
> In addition, in light of the reviewer’s suggestion, we will add an experiment showing that by increasing $m$ and $l$, the **sample variance reduces**.
>
> While the theoretical bounds may be looser in practice due to these dependencies, the empirical success of the ensemble strategy confirms the soundness of our approach.
>
> *W3. Q2. "This paper omits several strong modern baselines widely used... e.g., OpenCLIP ViT-bigG/EVA-CLIP, SigLIP families..."*
>
> **Response:** We agree that comparing against modern baselines is important to demonstrate robustness. We are currently finalizing experiments with **SigLIP** and **E5-V** and will include these in the revised paper and this thread.

---

> > ### Author Response · Authors · 2025-11-21
> >
> > *W4. "In the evaluations, the hard set definition uses CLIP AP < 0.5 to decide difficulty..."*
> >
> > **Response:** we would like to clarify that we report performance on this "hard set" for *all* baselines (Table 1), demonstrating that Needle outperforms not only CLIP but also ALIGN, FLAVA, and others on these difficult queries. Second, this methodology for defining query difficulty is not arbitrary; we adopted it from SeeSaw [1], a standard reference in interactive image search. This diagnostic choice allows us to specifically test whether our method solves the known limitations of the state-of-the-art.
> >
> > *W5. "having the Related Work in the supplementary material is a big downside..."*
> >
> > **Response:** We agree with the reviewer regarding the placement of the Related Work section, we had to move the related work to the Supplementary materials due to space constraints. We will move a summary of Appendix B to the main body of the manuscript in the revised manuscript.
> >
> > *Q3. "How sensitive are results to prompt phrasing and seed selection for the guide images?"*
> >
> > **Response:**
> > - **Seed Robustness:** As shown in our ablation study (Figure 4), performance stabilizes as the number of guide images (m) increases. By averaging over m generated images (Monte Carlo integration), we inherently smooth out the noise from specific random seeds.
> > - **Prompt Sensitivity:** While we use the raw query as the prompt, we have found that the current available foundation models are very robust to the input prompts, prior to implementing our final system, we tested various prompt cleaning strategies (e.g., removing parentheses and underscores) but we observed negligible difference compared to original prompt, which shows that usually foundation models are robust to minor changes in the prompt. We acknowledge that an interesting direction for future work would be using LLMs to semantically expand and detail the prompts to generate higher-fidelity guide tuples.
> >
> > [1] Moll, Oscar, et al. "SeeSaw: interactive ad-hoc search over image databases." Proceedings of the ACM on Management of Data 1.4 (2023): 1-26.

---

### Official Review · Reviewer_ZdRn · 2025-11-02

**Soundness:** 4
**Presentation:** 4
**Contribution:** 4
**Rating:** 8
**Confidence:** 4

**Summary:**

To improve the accuracy and speed of text-based image retrieval, the authors propose a system called NEEDLE. Instead of relying on contrastive learning to match text queries to images, NEEDLE re-formulates the task as an image-to-image search problem by using image foundation models to generate "guide images" based on the query text, which are then used to conduct the actual retrieval. The authors propose and integrate several practical optimizations to transform NEEDLE into a robust and efficient system, which results in low-latency (0.203s) retrieval on a 2xTesla T4 GPU machine, a latency competitive with that of non-generative methods such as a CLIP search. Quantitative benchmark numbers and a 20-user user study show that NEEDLE achieves state-of-the-art performance on text-based image retrieval compared to concurrent baselines.

**Strengths:**

Very solid baselines to compare against! Baselines include (in L239): CLIP [58], ALIGN [28]; FLAVA [65] and CoCa [85]; BLIP and MiniLM [39, 77]; and PlugIR [34].

A wide range of datasets for evaluating retrieval performance (L249)!

Very solid ablations that include the choice of hyperparameters (e.g. number of guide images and embedders in FIgure 4), choice of image generation models for creating guide images (Figure 5), and the separate effect of using generated guide images (Table 5).

The engineering seems very well thought-out to me. The paper has taken measures to improve the efficiency of the system for providing an actually good user experience, and not solely chasing benchmark numbers. The authors also provide details on the test system setup and include breakdowns for the inference time for running NEEDLE on LVIS, which I believe is fast enough for real-time user interactions with a handful of users despite the test system having somewhat outdated hardware (Tesla T4 GPUs).

It looks like the authors have made a fully-functional prototype for NEEDLE, which is great especially given the paper posits itself as making a "production-ready" system (L100).

User studies in Figure 12 shows NEEDLE is strongly favored over naive CLIP retrieval on LVIS.

**Weaknesses:**

Using Local Outlier Factor (LOF) to detect poor-quality generated guide images sounds like it would also eliminate any image that's "novel" or "unique" or "creative", because these would also cause a guide image to have a high LOF. If the user is an artist who draws distinct images (compared to the average internet images), and wants to search for one of their own creations among other more "nomral-looking" images, I believe the LOF rule would accidentally discard what may be valid guide images. Can the authors provide further justification for using LOF, or show that this unwanted elimination does not happen in practice? I'd be happy to raise my rating given further corroboration,.

**Questions:**

For Theorem 1 to hold, I think we also need some assumptions about the independence among the embeddings from different embedders to apply the Chernoff bound? That is, I think Theorem 1 should require $\{\mathcal{E}_l(\bar{g}_j)\}$ to be all independent. Yes, the embedders are distinctly trained, but the architectural and data biases would probably not make their embeddings of the same image independent. This may not cause a major issue in practice, but I believe the non-independence means the bound is less effective than Theorem 1 would suggest. It should not depend on $l$ as we cannot guarantee the embedders to give independent outputs.

I'm very interested in the discussion on "Reliance on Existing Models" (Appendix A). I wonder what new biases using generative image models introduces to image retrieval. An extended discussion with experiments would be appreciated.

Minor comments:
It'd be great to see more examples of the queries the users wrote during the user study. How complex are they? What "topics" did they talk about?

---

> ### Author Response · Authors · 2025-11-21
>
> We sincerely thank the reviewer for their support of our work and their feedback. We provide detailed responses to your insightful questions below.
>
> *W1. "Using Local Outlier Factor (LOF)... might accidentally discard what may be valid guide images (e.g., unique art)."*
>
> **Response:** We appreciate this keen observation regarding the potential risk of filtering out creative or novel guide images. First, we would like to clarify that the *sensitivity* of the outlier detection is controlled by the hyperparameter $\tau$, which allows users to adjust the strictness based on their specific domain needs (e.g., an artist searching for unique styles vs. a user searching for standard objects).
>
> Determining the optimal value for $\tau$ involves balancing a tradeoff inherent to our Generative Monte Carlo framework. On one hand, as the reviewer correctly mentioned, retaining outliers allows the model to capture *unique* relevant tuples; on the other, if a guide image is extremely distant from the other generated samples, it can *drag* the aggregated average away from the optimal expected value, which can drastically reduce the model's overall retrieval performance.
>
> In our practical experiments, we observed that for getting best results, the primary focus of the LOF mechanism should be on filtering out **erroneous tuples** like solid black images triggered by foundation model safeguards or complete noise, rather than stylistic variations. We found that even if a foundation model generates an approximate sketch or a stylistically unique image, it remains helpful for retrieval (it is in the acceptable vicinity w/o dragging the average far from the expected value). Therefore, the default configuration is tuned to target these "technical failures" rather than penalizing creativity. In light of the reviewer's insightful comment, we will add this justification to the revised manuscript.
>
> *Q1. "For Theorem 1 to hold... I think Theorem 1 should require embedders to be all independent."*
>
> **Response:** We agree with the reviewer that our analysis based on Chernoff bound in Theorem 1 assumes the independence of the embeddings, while this assumption may not be true in the real world.
> We realized that this issue has been studied in the classical sampling literature, given its popularity in real world settings.
>
> Particularly, following the classical effective sample size / design-effect analysis for equicorrelated observations [1,2], one can compute the effective sample size correction for equicorrelated observations.
>
> Specifically, if the pairwise correlation between the embeddings is $\rho$, the “effective number” of embedders is computed as:
>  $\ell_{eff} = \frac{\ell}{1+(\ell-1)\rho}$.
>
> As a result, by replacing $\ell$ by $\ell_{eff}$ in Theorem 1, we obtain a weaker bound for the correlated embeddings, as suggested by the reviewer. We will update the theorem, explicitly stating its assumptions, and will extend it for correlated embeddings.
>
> [1] Wolter, Kirk M., and Kirk M. Wolter. “Introduction to variance estimation”. Vol. 53. New York: Springer, 2007.
>
> [2] Kish, Leslie. "Survey sampling”. 1965.
>
> *Q2. "I wonder what new biases using generative image models introduces to image retrieval."*
>
> **Response:** This is a very interesting direction for discussion. In our experiments, we observed that foundation models introduce specific biases related to **compositionality** and **abstraction**. First, current foundation models often show a *"bag-of-words"* bias, to elaborate more, they usually fail to correctly depict specific compositional orderings (e.g., placing object A strictly to the left of object B), which propagates into the retrieval results. Second, we noted challenges with highly abstract concepts (e.g., "a sad day"). Foundation models tend to converge on specific visual attributes for these concepts (e.g., grayscale filters or rain), which creates a bias toward those specific visual features, potentially missing relevant images that depict "sadness" through different visual metaphors. In the light of reviewers observation, we are currently designing further experiments to identify these limitations and will expand Appendix A to include these preliminary findings.
>
> Finally, to mitigate individual foundation model impact, we would like to point out that our Monte-Carlo approach, allows us to use multiple **diverse** foundation models for querying, instead of relying only on one. Hence, we can rely on this collective wisdom of foundation models which will reduce the impact of individual model's failures and biases.
>
> *Q3. "It'd be great to see more examples of the queries the users wrote during the user study."*
>
> **Response:** We are happy to share some examples of user queries from our study:
> - El clasico in the Stadium
> - A red Subaru parked next to a white Camry
> - A PhD student defending his dissertation
>
> We will include a more comprehensive list of these queries in the supplementary material of the revised paper.

---

> > ### Comment · Reviewer_ZdRn · 2025-11-27
> >
> > The rebuttal has addressed my concerns and questions. I wish to maintain my original ratings.

---

### Official Review · Reviewer_JAPY · 2025-11-03

**Soundness:** 2
**Presentation:** 2
**Contribution:** 3
**Rating:** 4
**Confidence:** 4

**Summary:**

This paper introduces NEEDLE, a generative-AI-powered multimodal retrieval system designed to answer complex natural language queries over image datasets. Instead of relying on contrastive joint embeddings, the authors propose a Generative Monte Carlo framework that transforms a text query into multiple synthetic “guide” images using foundation models. These generated images are then embedded with multiple vision encoders (EVA, RegNet, etc.) and used to perform image-to-image nearest-neighbor search. The paper presents both a theoretical justification for Monte Carlo sampling, a system-level implementation with dynamic embedder trust weighting and outlier filtering, and an optimized inference pipeline that includes a query complexity classifier to bypass unnecessary image generation. Extensive experiments on benchmarks show consistent gains over prior vision–language models in retrieval metrics.

**Strengths:**

-  The idea of translating text queries into synthetic multimodal representations before retrieval is conceptually interesting. It reframes the retrieval task as a generative sampling problem, bridging generation and search.
- The paper goes beyond an algorithmic prototype and implements a deployable system with modular embedders, anomaly filtering, and caching. The inclusion of practical optimizations shows strong engineering effort.
- Comprehensive experiments are provided across multiple datasets with ablations on guide-image count, number of embedders, and foundation models. This gives a good sense of robustness under different configurations.

**Weaknesses:**

- The paper only compares against 2023-era contrastive models (CLIP, ALIGN, FLAVA, CoCa, BLIP). Recent multimodal embeddings such as UniIR[1] and E5-V[2] and VLM2Vec[3] using MLLM are now standard baselines for multimodal retrieval. Without evaluating against these stronger MLLM representations, it is unclear whether NEEDLE remains competitive in the modern multimodal landscape.

- The paper reports a total inference time of 0.203 s comparable to CLIP’s 0.184 s despite including computationally heavy image generation and multi-embedder inference. Hardware details, batch size, image resolution, and generator configuration are missing, making this efficiency claim scientifically unreliable.

- The evaluated benchmarks (COCO, LVIS, COLA, Winoground) test static image retrieval and compositional captions, not complex multimodal or reasoning-based queries. The claim of “natural-language query answering over multimodal databases” is overstated; the method is still restricted to image retrieval.

- The evaluation focuses only on static image retrieval tasks (COCO, LVIS, Winoground). This evidence is insufficient to support the paper’s claim of handling “complex natural-language queries.” If the authors truly intend to demonstrate compositional multimodal reasoning, a retrieval-augmented generation (RAG)–based visual question answering or multimodal reasoning setup would be far more appropriate. As it stands, the current retrieval-only setup does not convincingly show such complexity, and it is doubtful that the system would handle open-domain or knowledge-grounded VQA (e.g., Wikipedia-style) questions, where reasoning requires world knowledge.

- NEEDLE depends on multiple vision encoders, yet the paper provides no standalone evaluation of each embedder’s retrieval performance, parameter count, or computational footprint.

[1] Uniir: Training and benchmarking universal multimodal information retrievers. Wei et. al, ECCV24
[2] E5-v: Universal embeddings with multimodal large language models, Jiang et. al.,
[3] Vlm2vec: Training vision-language models for massive multimodal embedding tasks, Jiang et. al.,

**Questions:**

- Recent MLLM-based embedding models such as E5-V and VLM2Vec achieve much stronger performance and already support complex multimodal query understanding through instruction tuning. Have the authors compared NEEDLE against these newer baselines?

- The reported inference latency (0.203 s) is nearly identical to CLIP despite including foundation-model image generation and multiple encoders. Could the authors clarify the exact measurement conditions such as hardware, batch size, resolution, and generation model used?

- NEEDLE relies on encoders such as EVA and RegNet, but their individual parameter sizes, FLOPs, and retrieval accuracy are not reported. Could the authors provide quantitative comparisons that isolate the extent to which the proposed generative sampling contributes to the performance gain compared to embedders?

---

> ### Author Response · Authors · 2025-11-21
>
> We thank the reviewer for their constructive feedback and for recognizing the engineering effort behind Needle. We appreciate the opportunity to clarify our efficiency claims and experimental scope. We provide detailed responses to the specific concerns below.
>
> *W1. Q1. "The paper only compares against 2023-era contrastive models... Recent multimodal embeddings such as UniIR, E5-V, and VLM2Vec are now standard baselines...".*
>
> **Response:** We thank the reviewer for pointing out these recent and relevant baselines. We acknowledge that the field of multimodal retrieval is evolving rapidly. Our original selection of baselines was designed to cover the primary paradigms in zero-shot retrieval: Contrastive (CLIP, ALIGN), Unified (FLAVA, CoCa), Decoupled (BLIP+MiniLM), and LLM-based (PlugIR). However, we agree that a direct comparison with modern MLLM-based embeddings is beneficial to demonstrate Needle’s competitiveness in the current landscape. We are currently conducting experiments to include **E5-V** (as suggested) and **SigLIP** as additional baselines. We will add these results to the revised version of the paper to provide a comprehensive comparison against state-of-the-art MLLM representations.
>
> *W2. Q2. "The reported inference latency (0.203s) is nearly identical to CLIP... making this efficiency claim scientifically unreliable...".*
>
> **Response:** The 0.203s latency reported in Table 3 is accurate for our **optimized pipeline**. We apologize if the specific configuration details were not prominent enough; we will make them **more explicit** in the revised version. The measurements were taken on a server with two Tesla T4 GPUs. The optimized pipeline uses SD-Turbo as foundation model and generates a minimal configuration of guide images (1 image at 512x512 resolution), which our ablation study (Figure 4) proved is sufficient for high accuracy. As shown in the breakdown in Table 3, the image generation step takes only 0.136s using this configuration. Furthermore, as noted in Section 4, this latency represents the *"worst-case"* for complex queries. Our Query Complexity Classifier allows the system to short-circuit this pipeline for simple queries, achieving even lower average latencies. For the CLIP baseline, all experiments were run on the same hardware configuration, and we used the standard clip-vit-base-patch32 model (https://huggingface.co/openai/clip-vit-base-patch32)
>
> *W3. W4. "The evaluation focuses only on static image retrieval tasks... insufficient to support the paper’s claim of handling 'complex natural-language queries.'...".*
>
> **Response:** We would like to clarify that in the context of a **multimodal database system**, "query answering" refers to retrieving the correct tuple (image) that satisfies a user's constraint, rather than generating a textual answer (as in Visual Question Answering). Our definition of "complex queries" refers to retrieval tasks involving *compositionality, detailed descriptions*, and *spatial relationships* (e.g., "A blue napkin to the right of a white plate" ), which traditional dense retrievers (like CLIP) struggle with. The benchmarks we utilized, Winoground, COLA, and NoCaps, are specifically designed to test this type of compositional reasoning. While we agree that RAG/VQA are interesting directions, they represent a different problem statement (text generation) than the one addressed in this paper (database retrieval). We will refine our terminology to ensure the distinction between "complex retrieval queries" and "open-ended reasoning" is clear.
>
> *W5. Q3. "NEEDLE relies on encoders such as EVA and RegNet... provide quantitative comparisons that isolate the extent to which the proposed generative sampling contributes...".*
>
> **Response:** We actually provide this exact isolation in **Table 5 ("Contribution Analysis")**. To disentangle the gain of the generative method from the gain of using better embedders, we compared a "CLIP Baseline" against "Needle (CLIP-only)." In this experiment, Needle used CLIP as its only embedder but still employed the generative Monte Carlo framework. The results show that the generative method alone provided a **35% relative gain** in MAP on the LVIS dataset (0.168 to 0.228). This confirms that the performance boost is primarily driven by the Generative Monte Carlo approach, independent of the specific vision encoders used. The subsequent move to stronger embedders (EVA/RegNet) provides further, but separate, incremental gains. We will emphasize this finding in the text to better highlight the independent contribution of our proposed framework.

---

### Official Review · Reviewer_4JMw · 2025-11-04

**Soundness:** 3
**Presentation:** 3
**Contribution:** 2
**Rating:** 4
**Confidence:** 3

**Summary:**

This paper proposes using image generation model to perform Monte Carlo sampling, transforming complex natural language queries into multiple image queries for conducting complex natural language-to-image retrieval. Based on this framework, the paper designs a series of optimization methods to improve the efficiency of the overall system workflow. In the experiments, the proposed method outperforms the compared multimodal alignment approaches, particularly on the hard set.

**Strengths:**

The paper is logically well-structured. It is progressing smoothly from problem analysis, motivation, to the proposed solution, and then to optimization strategies. The approach is coherent and reasonable.
From a system perspective, this paper addresses and optimizes practical issues faced by the generate-and-retrieve paradigm, tackling problems of notable significance.
The experimental analysis is sufficiently thorough, making the results convincing.
The code framework is well-opened and complete, making deployment very convenient.

**Weaknesses:**

Using image generation models for cross-modal retrieval is not novel, as many related works [1,2,3] have conducted similar studies.
Although the paper emphasizes the use of foundation models, it only experiments with image generation models and does not include VLM, which intuitively might be more suitable for text-image retrieval tasks.
As Figure 5(d), using image generation model is significantly less efficient compared to general retrieval methods. Although the paper proposes several optimization strategies, the performance gap remains substantial.
[1] Zijun Long, et al. Diffusion Augmented Retrieval: A Training-Free Approach to Interactive Text-to-Image Retrieval.
[2] Ran Zuo, et al. SceneDiff: Generative Scene-Level Image Retrieval with Diffusion Models.
[3] Lan Wang, et al. Generative Zero-Shot Composed Image Retrieval.

**Questions:**

Since the method already employs a closed-source image generation model, it might be worthwhile to compare against more recent and stronger baselines, such as VLM-based embedding models. I am curious about the advantages of using an image generation model when compared with other methods of similar scale and complexity.

---

> ### Author Response · Authors · 2025-11-21
>
> We thank the reviewer for their insightful feedback. Several interesting questions were raised regarding novelty, baselines, and efficiency; we provide detailed responses below.
>
>
> *W1. "Using image generation models for cross-modal retrieval is not novel...".*
>
> **Response:** We thank the reviewer for highlighting this recent and relevant body of work [1, 2, 3]. This is a very active area of research, and we agree these are important comparisons. An earlier version of our paper was made publicly available **prior** to the publication of the works cited. While we cannot link to this preprint here to maintain anonymity, this timing establishes that these recent papers are either concurrent with or subsequent to our work.
>
> We acknowledge that this is a rapidly evolving field. In light of this concurrent literature, we will add **a remarks section** and **refine our novelty statement**. We will update our claim to be more precise, clearly distinguishing Needle's specific contributions, such as our Generative-based Monte Carlo framework and practical optimizations in system design, from these related approaches.
>
>
>
> *W2. Q1. "It might be worthwhile to compare against more recent and stronger baselines... I am curious about the advantages of using an image generation model...".*
>
> **Response:** We thank the reviewer for the suggestion. It is challenging to include every new model as the field evolves rapidly. To ensure a principled comparison, we initially categorized baselines into contrastive learning (CLIP, ALIGN) , unified models (FLAVA, CoCa) , decoupled pipelines (BLIP + MiniLM) , and LLM-based frameworks (PlugIR). We agree that a direct comparison to modern VLM-based embedding models is valuable. We are currently conducting experiments to include two SOTA VLM-based baselines, **SigLIP** and **E5-V**, and will add these results to the revised version of the paper and next comment in this thread.
>
>
> *W3. “Using image generation model is significantly less efficient... the performance gap remains substantial."*
>
> **Response:** This is a valid concern, and we dedicated Section 4 to addressing it. It is important to clarify that the 0.203s latency reported in Table 3 represents the *"worst-case"* time for complex queries that require the full generation pipeline. While this is higher than simple text-embedding lookups, it represents a significant improvement over our initial prototype (~5s) and is achieved through the optimizations detailed in the paper.
>
> Our primary strategy is to *avoid* this bottleneck for average-case queries. We introduced a Query Complexity Classifier that "short-circuits" the pipeline for simple queries, bypassing image generation entirely. As shown in Figure 5d, this allows Needle to achieve average inference speeds that are highly competitive with baselines like CLIP, while maintaining superior accuracy. In addition, our system improves this module “#hits” over time. The Implicit Metadata Generation module tags high-confidence results from complex queries with input query. This enriches the metadata over time, allowing the classifier to handle more queries as simple text-based searches, progressively reducing reliance on the generation pipeline.
>
> Finally, regarding the trade-off between speed and accuracy, we acknowledge that an image generation step is computationally intensive. Our goal was not to *beat* standard baselines on runtime, but to prove that for a reasonable and **acceptable production-level** efficiency cost, we can deliver higher retrieval accuracy, as demonstrated in our experimental evaluation. We believe this trade-off makes Needle a powerful and practical solution. To demonstrate this, we have released Needle as a fully open-source, production-ready system, installable on Linux and Mac.
>
> To summarize, Needle trades off a small increase in query time (but still interactive system) with significant improvement in accuracy.

---

### Author Response · Authors · 2025-11-27
**Summary of Changes**

**Summary of Revisions**

We sincerely thank all reviewers (**4JMw, JAPY, ZdRn, yRLh, aWQV**) for their time and constructive feedback. Your insights have helped us improve the quality, clarity, and theoretical depth of our paper. We have uploaded a revised manuscript incorporating the following key changes:

**1. Integration of New SOTA Baselines (Reviewers 4JMw, JAPY, yRLh)**
To address the fast-moving landscape of multimodal retrieval, we have added experimental results for two modern MLLM-based embedding models: **SigLIP** and **E5-V** (Table 1). These results demonstrate that Needle remains highly competitive and achieves superior performance on hard queries even against these latest advancements. Also, note that even these baselines can be a part of Needle ensemble in theory and further improve Needle's performance.

**2. Theoretical Refinements and Variance Analysis (Reviewers ZdRn, yRLh)**
* **Correlated Embedders:** Following **Reviewer ZdRn**’s insightful comment regarding the independence assumption in Theorem 1, we have updated the theoretical analysis to explicitly account for correlations between embedders using the "effective sample size" formulation $\ell_{eff} = \frac{l}{1+(l-1)\rho}$.
* **Estimator Variance:** Addressing **Reviewer yRLh**’s concern regarding estimator bias, we added a new ablation study (Figure 5) analyzing the sample variance. The results empirically validate that increasing the number of guide images ($m$) and embedders ($l$) consistently reduces variance, confirming the robustness of our Monte Carlo framework.

**3. Qualitative User Study Examples (Reviewer ZdRn)**
To provide better insight into the complexity of real-world usage, we have added **Table 7** in Appendix H.5 (Page 31). This table lists specific examples of user-generated queries from our study, ranging from simple objects (e.g., *"Celery stew"*) to complex scenes (e.g., *"A PhD student defending his dissertation"*).

**4. Sensitivity Analysis (Reviewer yRLh)**
We added a detailed discussion in **Appendix A** regarding the system's sensitivity to prompt phrasing and random seeds. This section clarifies the robustness of modern foundation models to minor syntactic variations in prompts.

**5. Related Work Placement and Novelty (Reviewers 4JMw, yRLh)**
We have moved a summary of the **Related Work** to the main body (Section 1) to better contextualize our contributions relative to concurrent generative retrieval methods. We also clarified our novelty claims, added the remarks sections and emphasized on Needle's novel contribution like being a *database system* that solves practical latency and robustness bottlenecks, rather than just an algorithmic approach.

We believe these revisions fully address the questions raised, and we are grateful for the opportunity to improve our work.

---

### Author Response · Authors · 2025-12-01
**Final Remarks: All Reviewer Concerns Resolved in Revision**

We thank the program organizers and the newly assigned Area Chair for their time and effort in managing the review process under these exceptional circumstances. In the revised manuscript, we fully addressed the concerns raised by the reviewers. The following is the summary of changes:

- **Modern Baselines Added:**
We integrated SigLIP and E5-V into our experiments (Table 1). Needle continues to outperform these state-of-the-art MLLM embeddings, proving our method remains competitive to recent SOTA architectures.

- **Novelty & Related Works:**
We distinguished Needle as a Database System solving practical deployment bottlenecks (latency, outliers), differentiating it from concurrent works, we also added "novelty remark" and moved a summary of related works to the revised manuscript.

- **Theoretical Refinements and Variance Analysis:**
We updated our theoretical analysis to explicitly account for correlated embedders using an "effective sample size" formulation. Additionally, we added a new ablation study (Figure 5) which empirically validates that increasing the number of guide images and embedders reduces sample variance, confirming the robustness of the Monte Carlo framework.

- **Minor Changes:**
(a) We clarified the specific configuration (SD-Turbo) that achieves 0.203s latency, comparable to CLIP, and demonstrated how our Query Complexity Classifier minimizes overhead. (b) To provide insight into real-world usage complexity, we added specific user query examples, to the case study in Appendix H.5. (c) we also added a sensitivity analysis in Appendix A to discuss the system's robustness regarding prompt phrasing and random seeds.

We thank the reviewers for their constructive feedback. Having addressed the reviewers' comments, as confirmed by Reviewer ZdRn, we believe that the revised manuscript is now in solid shape.

---

### Meta-Review · Area_Chair_hTyo · 2026-01-07

**Summary:**

This paper proposes Needle, an image retrieval system that models complex query semantics through synthetic data generation rather than contrastive learning or metadata-based retrieval. Needle has shown a strong extensibility, which can integrate different foundation models with embedders. Extensive experiments on benchmark datasets demonstrate that our system significantly outperforms existing text-to-image retrieval methods.

**Reviewer Concerns:**

1. The paper does not include comparisons with several representative state-of-the-art models. During the rebuttal, the authors conducted additional experiments, including SigLIP and E5-v. The results indicate that while Needle does not outperform E5-v, it achieves comparable performance. However, this limits the strength of the empirical claims regarding superiority.
2. The motivation of the paper remains somewhat unclear. Using image generation models for cross-modal retrieval is not a novel idea. In the rebuttal, the authors attempted to clarify the novelty by emphasizing Needle's specific contributions, such as the generative-based Monte Carlo framework and practical system-level optimizations. Nevertheless, some experimental settings and results are insufficiently described, and the provided evidence does not fully substantiate these claimed contributions.
3. To better support the proposed claims, experiments on a broader range of datasets are necessary to demonstrate the robustness and general effectiveness of the method.
4. The related work is not sufficiently comprehensive and should be included in the main body of the paper rather than being placed in the appendix.

**Reviewer Scores:**

The authors have attempted to address the reviewers' concerns. However, the method does not demonstrate clear performance improvements over existing VLM-based models, and the motivation remains insufficiently clarified. As a result, these issues may raise further concerns, and the overall assessment score is unlikely to change.

---

### Decision · Program_Chairs · 2026-01-26

Reject